# In silico re-engineering of a neurotransmitter to activate KCNQ potassium channels in an isoform-specific manner

Rían W. Manville[1] & Geoffrey W. Abbott[1]*

Voltage-gated potassium (Kv) channel dysfunction causes a variety of inherited disorders, but developing small molecules that activate Kv channels has proven challenging. We recently discovered that the inhibitory neurotransmitter γ-aminobutyric acid (GABA) directly activates Kv channels KCNQ3 and KCNQ5. Here, finding that inhibitory neurotransmitter glycine does not activate KCNQs, we re-engineered it in silico to introduce predicted KCNQ-opening properties, screened by in silico docking, then validated the hits in vitro. Attaching a fluorophenyl ring to glycine optimized its electrostatic potential, converting it to a low-nM affinity KCNQ channel activator. Repositioning the phenyl ring fluorine and/or adding a methylsulfonyl group increased the efficacy of the re-engineered glycines and switched their target KCNQs. Combining KCNQ2- and KCNQ3-specific glycine derivatives synergistically potentiated KCNQ2/3 activation by exploiting heteromeric channel composition. Thus, in silico optimization and docking, combined with functional screening of only three compounds, facilitated re-engineering of glycine to develop several potent KCNQ activators.

[1] Bioelectricity Laboratory, Department of Physiology and Biophysics, School of Medicine, University of California, Irvine, CA, USA. *email: abbottg@uci.edu

Voltage-gated potassium (Kv) channel pore-forming α subunits are generated by a numerous and diverse gene family comprising 40 members in the human genome, separated into 12 subfamilies. Native Kv channels also contain regulatory subunits that shape their functional properties and further expand their diversity and functional repertoire. Kv channels are essential for a wide range of physiological processes, and in many cases little functional redundancy is observed even between seemingly closely related isoforms. Accordingly, disruption of specific Kv channel α or β subunits by inherited or sporadic human gene variants (or gene deletion in mice) is associated with a variety of disease syndromes, many of which are severe and often lethal[1].

Despite in-depth knowledge of many of the physiological functions of specific Kv channels, and of the pathophysiological consequences of their disruption, therapeutic pharmacological targeting of the channels has been challenging. One of the reasons for this is that a great many Kv channel-linked disorders, or channelopathies, arise from loss of function. Direct correction of these requires, therefore, channel openers—a more difficult task than developing channel inhibitors or blockers.

One of the best known Kv channel openers is the drug retigabine (ezogabine). Retigabine activates neuronal KCNQ channels by negative-shifting their voltage dependence of activation[2,3]. Heteromeric KCNQ2/3 channels are particularly important in generating the muscarinic-inhibited M-current, a background Kv current that acts as a gatekeeper to limit aberrant neuronal firing[4,5]. Retigabine was the first Kv channel opener to reach the clinic, but was withdrawn in 2017 because of off-target side effects—it turns the sclera and skin blue[6]. However, it is effective at opening KCNQ2/3 channels and was clinically useful, mostly as an add-on therapy, in epilepsy. Since the development of retigabine, a new syndrome was recognized, termed KCNQ2 encephalopathy[7]. Caused primarily by sporadic, KCNQ2 loss-of-function mutations (as carriers tend not to reproduce), this disease is notable for severe developmental delays in addition to epilepsy. Clearly, new activators of KCNQ2 and many other Kv channels are needed, and this need will be acknowledged further as other Kv channelopathies are identified.

We recently made the surprising discovery that the predominant inhibitory neurotransmitter γ-aminobutyric acid (GABA) binds in a similar binding pocket to that of retigabine, and activates KCNQ3, KCNQ5, and KCNQ2/3 channels[8]. We also found that other metabolites, GABA analogs, and phytochemicals bind to a similar site, the majority also opening KCNQ channels by favoring their activation at more hyperpolarized membrane potentials[8–10]. KCNQ channels, and possibly other Kv channels, thereby possess a binding pocket that accommodates numerous types of small-molecule activators.

Glycine, which is structurally related to GABA, is also an inhibitory neurotransmitter. Here, after finding that glycine does not activate KCNQ channels, we re-engineered the glycine structure in silico to introduce known properties of KCNQ activators, and tested candidates using docking simulations. With minimal real-world functional screening, this led to discovery of a series of potent KCNQ channel openers, including a pair of activators that leverage isoform preferences to synergistically activate KCNQ2/3.

## Results

### In silico re-engineering glycine to activate KCNQ channels.
Synthetic anticonvulsants such as retigabine possess negative electrostatic surface potential near their carbonyl groups, a property found to be important for their activation of KCNQ2/3 channels[11] (Fig. 1a). GABA possesses this same chemical property and also activates KCNQ2/3 channels[8] (Fig. 1b). Here, we show that glycine, the next most prominent inhibitory neurotransmitter, exhibits relatively weaker negative electrostatic surface potential that is not well centered at its carbonyl oxygen (Fig. 1c), nor does it activate KCNQ2/3, unlike GABA and retigabine, which activated KCNQ2/3 channels here with $EC_{50}$ values of $220 \pm 160$ nM and $6.9 \pm 0.34$ µM, respectively (Fig. 1d, e) (Supplementary Data 1, Tables 1–3). We hypothesized that re-engineering glycine to center surface negative electrostatic potential on its carbonyl oxygen would endow it with the capability to open KCNQ2/3. We therefore next mapped in silico the surface charge of a number of glycine derivatives.

In the simplest derivatives, surface negative potential was still skewed away from the carbonyl (Fig. 1f). In several more complex structures, including iminodiacetic acid glycine derivatives and N-[Bis(methylthiomethylene]glycine, the negative potential was skewed and/or partially hidden; in N,N-Bis(phosphonomethyl) glycine the negative potential was centered around a phosphate oxygen rather than the glycine carbonyl oxygen (Fig. 1g). However, replacing one of the glycine amino group hydrogens with a substituted phenyl ring centered a strong negative surface potential on the glycine carbonyl oxygen; 4-(fluorophenyl)glycine (4FPG) resulted in a single center of electronegative surface charge, while 4-(hydroxyphenyl)glycine also exhibited a second center of electronegative surface potential at the phenyl hydroxyl group (Fig. 1h).

For the second in silico prediction phase, using SwissDock we performed unbiased docking prediction analysis, of the glycine derivatives to a KCNQ1-KCNQ3 chimeric model[8] based on the recent cryo-EM derived KCNQ1 structure[12]. We were especially interested in binding in the pocket lined on one side by the S5 tryptophan (W265 on KCNQ3) that is important for retigabine and GABA binding[8,13], and on the other side by the S4-S5 linker-proximal arginine at the foot of S4 (R242 in KCNQ3) that is required for binding of phytochemicals such as mallotoxin to KCNQ channels[14] and mutation of which in KCNQ2 causes benign familial neonatal convulsions[15] (Fig. 2a–c). As expected from its chemical properties and lack of effects on KCNQ2/3, glycine failed to dock (Fig. 2d). In contrast, 4FPG docked in the binding pocket (*red oval*, Fig. 2e) whereas other glycine derivatives did not (black ovals, Fig. 2e). The docking position of 4FPG was closer to the S4-5 arginine (purple) than to the S5 tryptophan (red) (Fig. 2f). Surface electrostatic potential plotting shows that 4FPG possesses negative charge (red) close to its carbonyl oxygen (Fig. 2g). In the majority of poses, 4FPG was positioned lengthways between the S5 W and the S4-5 R with either the fluorine or the carbonyl oxygen in 4FPG proximal to the S5 tryptophan (Fig. 2h).

### 4FPG isoform-selectively activates KCNQ channels.
We next validated the in silico predictions using two-electrode voltage clamp of homomeric neuronal KCNQ2-5 channel isoforms expressed in *Xenopus laevis* oocytes. By quantifying the hyperpolarizing shift in voltage dependence of KCNQ channel activation ($\Delta V_{0.5act}$) versus [4FPG], we discovered that, as predicted, 4FPG is a KCNQ channel opener. 4FPG most potently activated (i.e., negative-shifted the voltage dependence of activation of) KCNQ4 ($EC_{50} = 49 \pm 12$ nM), followed by KCNQ2 ($EC_{50} = 69 \pm 31$ nM) and KCNQ1 ($EC_{50} = 90 \pm 20$ nM), and had no effects on KCNQ3* or KCNQ5 (Fig. 2i–k) (Supplementary Data 1, Tables 4–8). As in previous studies[8,10], we used the A315T KCNQ3 mutant (KCNQ3*) that ensures large enough currents to accurately quantify voltage dependence and pharmacology of homomeric KCNQ3[16]. 4FPG speeded KCNQ2 and KCNQ4 activation and slowed their deactivation, suggesting that it opens

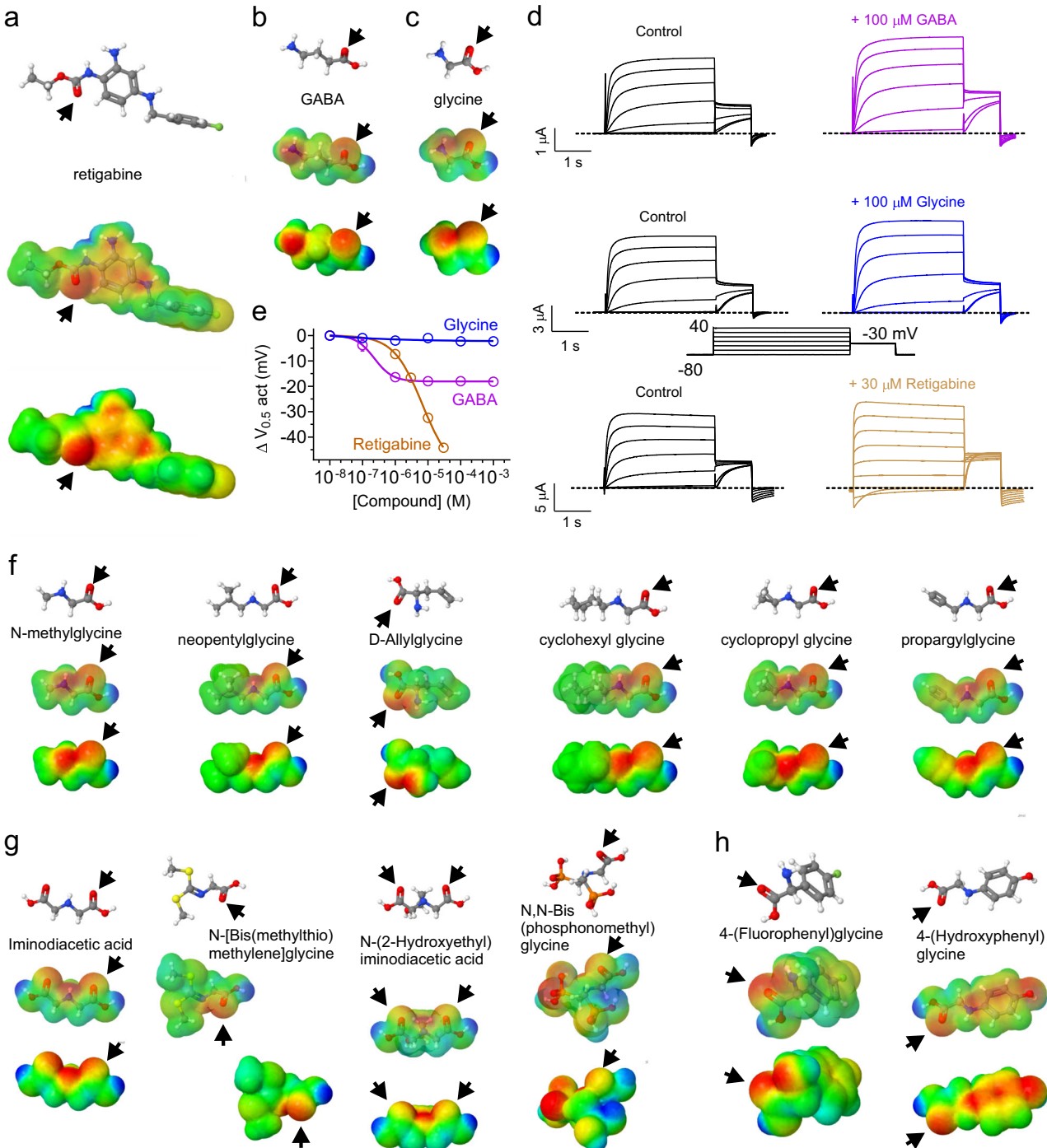

**Fig. 1** In silico engineering predicted KCNQ-opening properties into glycine. **a** Retigabine structure, electrostatic surface potentials (red, electron-dense; blue, electron-poor; green, neutral) and an overlay of the two, all calculated and plotted using Jmol. Arrow carbonyl oxygen. **b** GABA, parameters as in (**a**). **c** Glycine, parameters as in (**a**). **d** Mean traces showing effects of GABA, glycine and retigabine on KCNQ2/3 channels expressed in *Xenopus* oocytes ($n = 4$–6). Voltage protocol (inset) was used for all TEVC recordings in this study unless otherwise indicated. **e** KCNQ2/3 dose response to glycine, GABA and retigabine, quantified from recordings as in (**d**) as the shift in voltage dependence of activation ($\Delta V_{0.5act}$) measured from the tail current. Error bars indicate SEM; $n = 4$–6. **f** Structures and surface potential plots (as in (**a**)) for the simple glycine derivatives indicated; *arrows*, carbonyl oxygen. **g** Structures and surface potential plots (as in a) for the double-carbonyl or branched glycine derivatives indicated; arrows, carbonyl oxygen. **h** Structures and surface potential plots (as in (**a**)) for the glycine derivatives bearing a phenyl ring; arrows indicate carbonyl group

these channels by stabilizing an open state and/or destabilizing a closed state. In contrast, 4FPG slowed KCNQ1 deactivation but did not speed its activation, suggesting 4FPG may solely destabilize the closed state in KCNQ1 (Fig. 2k, l) (Supplementary Data 1, Tables 9–12). This is potentially of mechanistic interest, as KCNQ1 lacks the S5 W required for activation by retigabine, yet

its activity is still potentiated by 4FPG. The data suggest that the S5 W may be important for effects on the activated state, yet is not required for effects on the deactivated state, at least in KCNQ1. The preference for KCNQ4 activation was in contrast to GABA and gabapentin, which we previously found[8] to each activate only KCNQ3 and KCNQ5, and to retigabine which

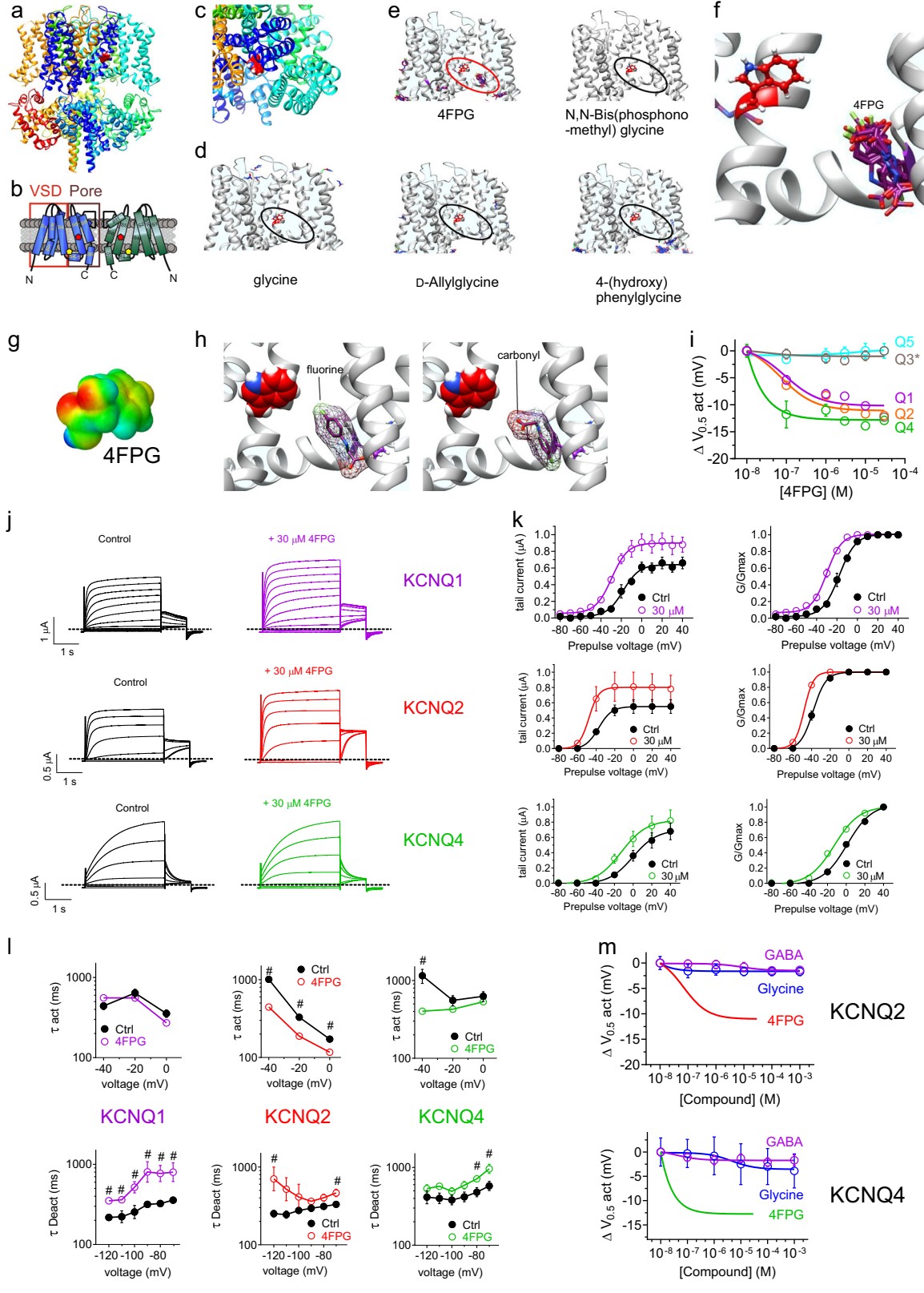

favors KCNQ3 and activates KCNQ2, KCNQ4 and KCNQ5 to a lesser extent, and does not activate KCNQ1[8,9,17]. Thus, in contrast to 4FPG, here glycine and GABA were unable to open KCNQ2 and KCNQ4 homomers (Fig. 2m) (Supplementary Data 1, Tables 13–16).

**Subtle modifications to 4FPG create derivatives with altered KCNQ isoform selectivity.** KCNQ1 and KCNQ4 are expressed in multiple tissues (KCNQ4 in the auditory system and vasculature, KCNQ1 in the cardiovascular system and multiple epithelia) and their activation might cause unwanted off-target effects if one

**Fig. 2** In silico prediction and in vitro validation of a KCNQ-activating glycine derivative. All error bars indicate SEM. **a** Chimeric KCNQ1/KCNQ3 structural model (red, KCNQ3-W265). **b** Topological representation of KCNQ channel showing two of the four subunits, without domain swapping for clarity. *Pentagon*, approximate position of KCNQ3-W265; VSD, voltage sensing domain. **c** Close-up extracellular view of KCNQ1/KCNQ3 structural model (red, KCNQ3-W265). **d** Docking result showing predicted lack of binding of glycine to the KCNQ1/KCNQ3 structural model. Red, KCNQ3-W265; black oval highlights lack of glycine binding in the typical binding zone for GABA and its metabolites and analogs. **e** Docking results for various glycine derivatives illustrated in Fig. 1 showing predicted binding of 4FPG within the GABA binding pocket (highlighted by red oval) but not of the other molecules (black ovals). All predicted binding configurations shown overlaid for each molecule. **f** Close-up of predicted binding poses of 4FPG within the GABA binding pocket. **g** Surface electrostatic potential plot of 4FPG. **h** Comparison of two different predicted orientations of 4FPG within the KCNQ binding pocket, as predicted by SwissDock. **i** 4FPG dose responses for homomeric KCNQ1, 2, 3*, 4, and 5 channels expressed in oocytes, quantified as shift in the voltage dependence of channel activation ($\Delta V_{0.5act}$); $n = 4$–6. **j** Mean traces showing effects of 4FPG (30 μM) on KCNQ1, KCNQ2 and KCNQ4; $n = 4$–6. **k** Effects of 4FPG (30 μM) on KCNQ1, KCNQ and KCN raw and normalized (G/Gmax) tail current, calculated from traces as in panel **j**; $n = 4$–6. **l** Effects of 4FPG (30 μM) on KCNQ1, KCNQ2, and KCNQ4 activation (act) and deactivation (Deact) rates, fitted as a single exponential function (τ); $n = 4$–6. **m** 4FPG dose responses for KCNQ2 and KCNQ4 compared to those of glycine and GABA, quantified as shift in voltage dependence of activation ($\Delta V_{0.5act}$); $n = 5$–6

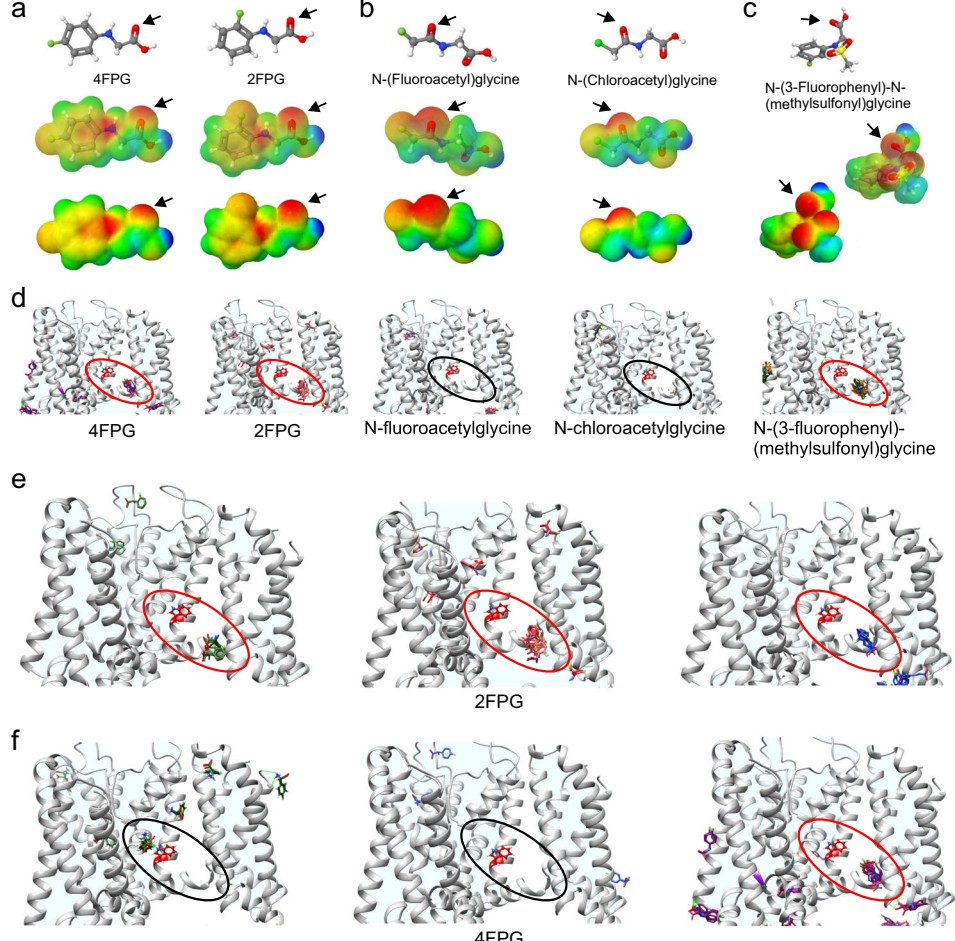

**Fig. 3** In silico prediction of 4FPG-related KCNQ-activating glycine derivatives. **a** Chemical properties of 4FPG versus 2FPG: structure, electrostatic surface potentials (red, electron-dense; blue, electron-poor; green, neutral) and an overlay of the two, all calculated and plotted using Jmol. Arrows, native glycine carbonyl. **b**, **c** Chemical properties of 4FPG-related glycine derivatives, parameters as in (**a**). Arrows, non-native glycine carbonyls for N-(fluoroacetyl)glycine and N-(Chloroacetyl)glycine; native glycine carbonyl for 3FMSG. **d** Docking results showing predicted binding (red ovals) or lack thereof (black ovals) of the compounds in (**a–c**) to the GABA binding pocket in the KCNQ1/KCNQ3 structural model. Red side-chain, KCNQ3-W265. **e** Docking results showing predicted binding (red ovals) of three different conformational forms of 2FPG to the GABA binding pocket in the KCNQ1/KCNQ3 structural model. Red side-chain, KCNQ3-W265. **f** Docking results showing predicted binding (red oval) or lack thereof (black ovals) of three different conformational forms of 4FPG to the GABA binding pocket in the KCNQ1/KCNQ3 structural model. Red side-chain, KCNQ3-W265

were instead intending to target KCNQ2 in epilepsy, for example. We therefore further re-engineered 4FPG in silico, creating several single-halide glycine derivatives, all of which exhibited negative electrostatic surface potential close to a carbonyl oxygen (Fig. 3a–c). Interestingly, only those that contained a

fluorophenyl ring concentrated negative surface potential at the native glycine carbonyl and were predicted to dock in the KCNQ1-KCNQ3 chimeric model. In addition to 4FPG these were 2-(fluorophenyl)glycine (2FPG) and N-(fluorophenyl)-N-(methylsulfonyl)glycine (3FMSG). Those that centered negative

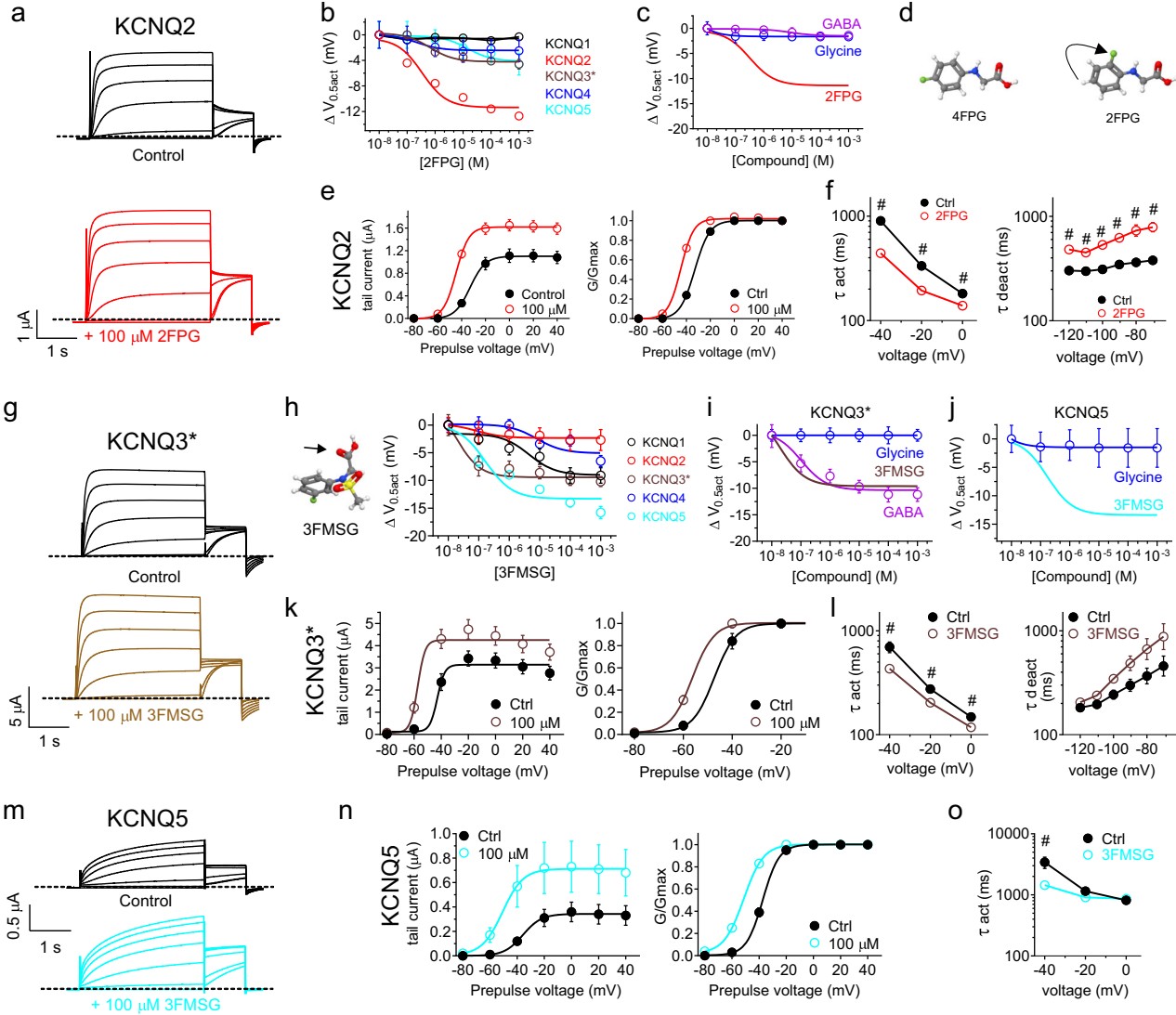

**Fig. 4** KCNQ isoform-specific activation by fluorinated glycine derivatives. All error bars indicate SEM. **a** Mean traces showing effects of 2FPG (100 µM) on KCNQ2 ($n = 5$). **b** 2FPG dose responses for homomeric KCNQ1, 2, 3*, 4, and 5, quantified as shift in the voltage dependence of channel activation ($\Delta V_{0.5act}$) calculated from the tail current using recordings as in panel (**a**); $n = 5$. **c** 2FPG dose response for KCNQ2 compared to those of glycine and GABA, quantified as current fold-change at −60 mV; $n = 5$–6. **d** Comparison of 4FPG and 2FPG structures showing the change in fluorine position (arrow). **e** Effects of 2FPG (100 µM) on KCNQ2 raw tail currents and normalized tail current (G/Gmax); $n = 5$. **f** Effects of 2FPG (100 µM) on KCNQ2 activation and deactivation rates, fitted as a single exponential function ($\tau$); $n = 5$. # $P < 0.01$. **g** Mean traces showing effects of 3FMSG (100 µM) on KCNQ3* ($n = 7$). **h** 3FMSG (glycine carbonyl highlighted with arrow) dose responses for homomeric KCNQ1, 2, 3*, 4, and 5 channels quantified as $\Delta V_{0.5act}$ measured from the tail currents from traces as in (**g**); $n = 4$–7. **i** 3FMSG dose response for KCNQ3* compared to those of glycine and GABA, quantified as $\Delta V_{0.5act}$; $n = 5$–7. **j** 3FMSG dose response for KCNQ5 compared to that of glycine, quantified as current fold-change at −60 mV; $n = 4$–5. **k** Effects of 3FMSG (100 µM) on KCNQ3* raw tail current and normalized tail current (G/Gmax); $n = 7$. **l** Effects of 3FMSG (100 µM) on KCNQ3* activation and deactivation rates, fitted as a single exponential function ($\tau$); $n = 7$. **m** Mean traces showing effects of 3FMSG (100 µM) on KCNQ5 ($n = 5$). **n** Effects of 3FMSG (100 µM) on KCNQ5 raw tail currents and normalized tail current (G/Gmax) measured from traces as in panel m; $n = 5$. **o** Effects of 3FMSG (100 µM) on KCNQ5 activation rate, fitted as a single exponential function ($\tau$); $n = 5$. # $P < 0.05$

surface potential at the halide-proximal carbonyl and lacked the phenyl group (N-(fluoroacetyl)glycine and N-(chloroacetyl)glycine) did not dock (Fig. 3d).

The most closely related glycine derivative to 4FPG was 2-(fluorophenyl)glycine (2FPG), the only difference being the fluorine position on the phenyl ring. 4FPG and 2FPG are each represented in several conformations on the Zinc database (http://zinc.docking.org/) used for ligand selection, and so we compared docking of all conformations. All three 2FPG conformations docked primarily to the known binding pocket (Fig. 3e) whereas 4FPG showed more docking heterogeneity and

two of three of its conformations did not dock in the neurotransmitter binding pocket (Fig. 3f). This suggested 4FPG and 2FPG might have different KCNQ opening characteristics, which we tested next.

Strikingly, 2FPG was a potent KCNQ2 isoform-selective opener (EC$_{50}$ = 322 ± 139 nM) with negligible effects on KCNQ1, 3*, 4, or 5 (Fig. 4a, b) (Supplementary Data 1, Tables 17–21). The KCNQ2 activation by 2FPG was in contrast to the non-KCNQ2 activating GABA (as we previously reported[8]) and glycine (Fig. 4c). The loss of KCNQ4 opening activity, in contrast to 4FPG, was remarkable given the small differences in the 2FPG

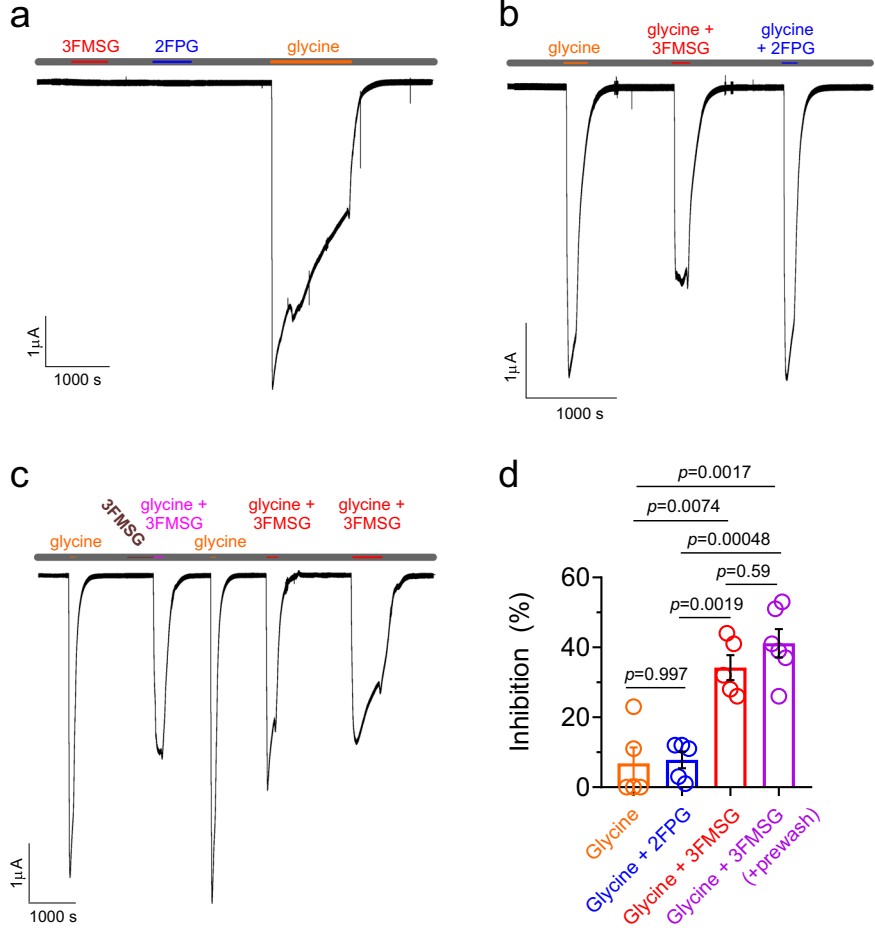

**Fig. 5** Differential effects on GLRA1 activity of 2FPG and 3FMSG. All error bars indicate SEM. **a** Exemplar trace showing lack of effects of 3FMSG or 2FPG alone on GLRA1 activty, compared to robust activation by glycine alone (all compounds applied at 1 mM), application indicated by colored bars at top. Gray, application of bath solution alone. **b** Exemplar trace showing inhibition of glycine-activated GLRA1 by 3FMSG (100 μM) but not 2FPG (100 μM) (glycine applied at 1 mM in each case), application indicated by colored bars at top. Gray, application of bath solution alone. **c** Exemplar trace showing inhibition of glycine-activated GLRA1 by 3FMSG (100 μM) with versus without 3FMSG (100 μM) pre-wash in (glycine applied at 1 mM in each case), application indicated by colored bars at top. Gray, application of bath solution alone. **d** Mean inhibition of 1 mM glycine-activated GLRA1 current by 2FPG or 3FMSG (100 μM) with/without 3FMSG (100 μM) pre-wash, n = 5–6, from traces as in panels (**b**) and (**c**). Currents were compared to an initial current activated by glycine alone; as a control, glycine-activated current in a subsequent wash-in was compared to the initial glycine wash-in current (orange), showing negligible inhibition as expected

and 4FPG structures (Fig. 4d). 2FPG negative-shifted the voltage dependence of KCNQ2 activation but also increased KCNQ2 currents at more depolarized potentials (Fig. 4b, e) and speeded KCNQ2 activation and slowed deactivation as for 4FPG (Fig. 4f) (Supplementary Data 1, Tables 22 and 23).

3FMSG exhibited still different selectivity, activating KCNQ3* most potently (EC$_{50}$ = 18 ± 12 nM) followed by KCNQ5 (EC$_{50}$ = 171 ± 112 nM), with negligible or much less potent effects on KCNQ1, KCNQ2 and KCNQ4 (Fig. 4g, h). In contrast, glycine had no effects on homomeric KCNQ3* or KCNQ5 (Fig. 4i, j) (Supplementary Data 1, Tables 24–31). 3FMSG negative-shifted the activation voltage dependence of KCNQ3* and also increased current at positive voltages (Fig. 4k), speeded activation and slowed deactivation (Fig. 4l). 3FMSG exerted similar effects on KCNQ5 and in addition was more effective at increasing KCNQ5 current compared to KCNQ3* current at positive potentials (Fig. 4m–o) (Supplementary Data 1, Tables 32–34).

**3FMSG inhibits glycine receptor GLRA1.** We next tested whether the glycine derivatives 2FPG and 3FMSG modulated the canonical glycine receptor, GLRA1. Neither compound activated

GLRA1 at 1 mM, in contrast to glycine (1 mM) (Fig. 5a). However, 3FMSG but not 2FPG (each at 100 μM) partially inhibited the activation of GLRA1 by 1 mM glycine (Fig. 5b). The degree of GLRA1 inhibition by 3FMSG was similar (~40%) whether or not 3FMSG alone was applied immediately before co-application of glycine with 3FMSG (Fig. 5c, d). Interestingly, co-application of 3FMSG also reduced the degree of desensitization observed during glycine activation of GLRA1 (Fig. 5b, c). Thus, 3FMSG but not 2FPG retains its ability to bind to a canonical glycine receptor, but acts as an inhibitor rather than an activator. The 40% inhibition by 3FMSG at 100 μM suggests a lower potency than its submicromolar activating effects on KCNQ3* and KCNQ5, although the comparison is difficult to quantify as 3FMSG may be competing with glycine, assuming it binds to the same site, on GLRA1.

**Glycine derivatives differentially activate KCNQ2/3 channels.** As KCNQ2/3 heteromers are the predominant neuronal KCNQ isoform, we tested their sensitivity to the glycine derivatives. All three derivatives activated KCNQ2/3 (Fig. 6a, b), with 2FPG and 3FMSG the more efficacious at opening KCNQ2/3 and at

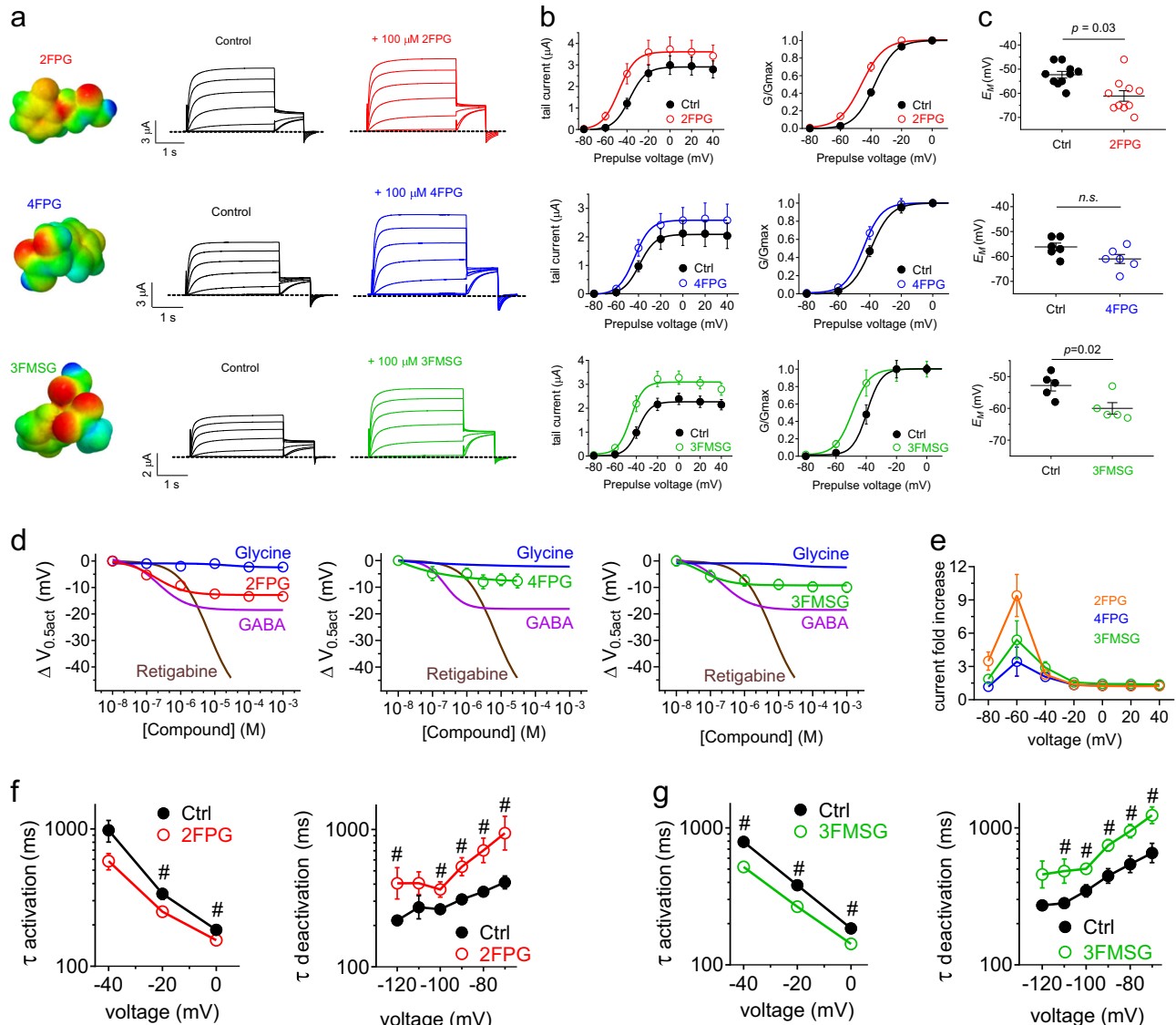

**Fig. 6 Differential effects on KCNQ2/3 activity of fluorinated glycine derivatives.** All error bars indicate SEM. **a** Jmol surface plot of compounds indicated, showing electrostatic surface potential (red, negative; blue, positive) and corresponding mean TEVC traces for KCNQ2/3 expressed in *Xenopus* oocytes in the absence (control) or presence of compounds as glycine derivatives as indicated (n = 4–6). Dashed lines indicated zero current level. **b** Mean tail current and normalized tail currents (G/Gmax) versus prepulse voltage relationships recorded by TEVC in *Xenopus* oocytes expressing KCNQ2/3 channels in the absence (black) or presence (red, blue, green) of glycine derivatives indicated (100 µM) (n = 4–6). **c** Effects of glycine derivatives (100 µM) on resting membrane potential ($E_M$) of unclamped oocytes expressing KCNQ2/3 (n = 4–6). **d** 2FPG, 4FPG and 3FMSG dose responses for KCNQ2/3 compared to those of glycine, GABA and retigabine, quantified as shift in voltage dependence of activation ($\Delta V_{0.5act}$); n = 4–6. **e** Comparison of 2FPG, 4FPG and 3FMSG (100 µM) effects quantified as KCNQ2/3 current fold-increase versus membrane potential; n = 4–6. **f** Effects of 2FPG (100 µM) on KCNQ2/3 activation and deactivation rates, fitted as a single exponential function ($\tau$); n = 15. # P < 0.05 between values at equivalent membrane potential. **g** Effects of 3FMSG (100 µM) on KCNQ2/3 activation and deactivation rates, fitted as a single exponential function ($\tau$); n = 10. # P < 0.05 between values at equivalent membrane potential

KCNQ2/3-dependently shifting the resting membrane potential (Fig. 6c). All three compounds were much more potent than retigabine (2FPG, $EC_{50} = 184 \pm 15$ nM; 3FMSG, $EC_{50} = 51 \pm 21$ nM; 4FPG, $EC_{50} = 61 \pm 42$ nM; retigabine, $EC_{50} = 6.87 \pm 0.34$ µM) although none came close to matching its efficacy upon KCNQ2/3 (Fig. 6d, e) (Supplementary Data 1, Tables 35–37). As we observed for homomeric channels, 2FPG and 3FMSG speeded KCNQ2/3 activation and slowed its deactivation (Fig. 6f, g) (Supplementary Data 1, Tables 38–41).

**2FPG and 3FMSG occupy the KCNQ neurotransmitter binding pocket.** Docking poses predicted that 2FPG binds between the

S5 tryptophan (W236 in KCNQ2) and the arginine at the foot of S4 (R213 in KCNQ2) (Fig. 7a, b). Wash-in and washout experiments revealed immediate onset of 2FPG effects on KCNQ2/3 channels upon commencing wash-in, and immediate reduction in current upon commencing washout; glycine as expected had no effect (Fig. 7c). Fitting with a single exponential function the effects on KCNQ2/3 current during 2FPG wash-in and washout revealed tau values of $16 \pm 3$ s and $42 \pm 9$ s, respectively (Fig. 7d). These time-courses are not compatible with 2FPG having to cross the plasma membrane to access its binding site from the intracellular face of the plasma membrane, which would typically not give rise to immediate onset of effects during wash-in and would

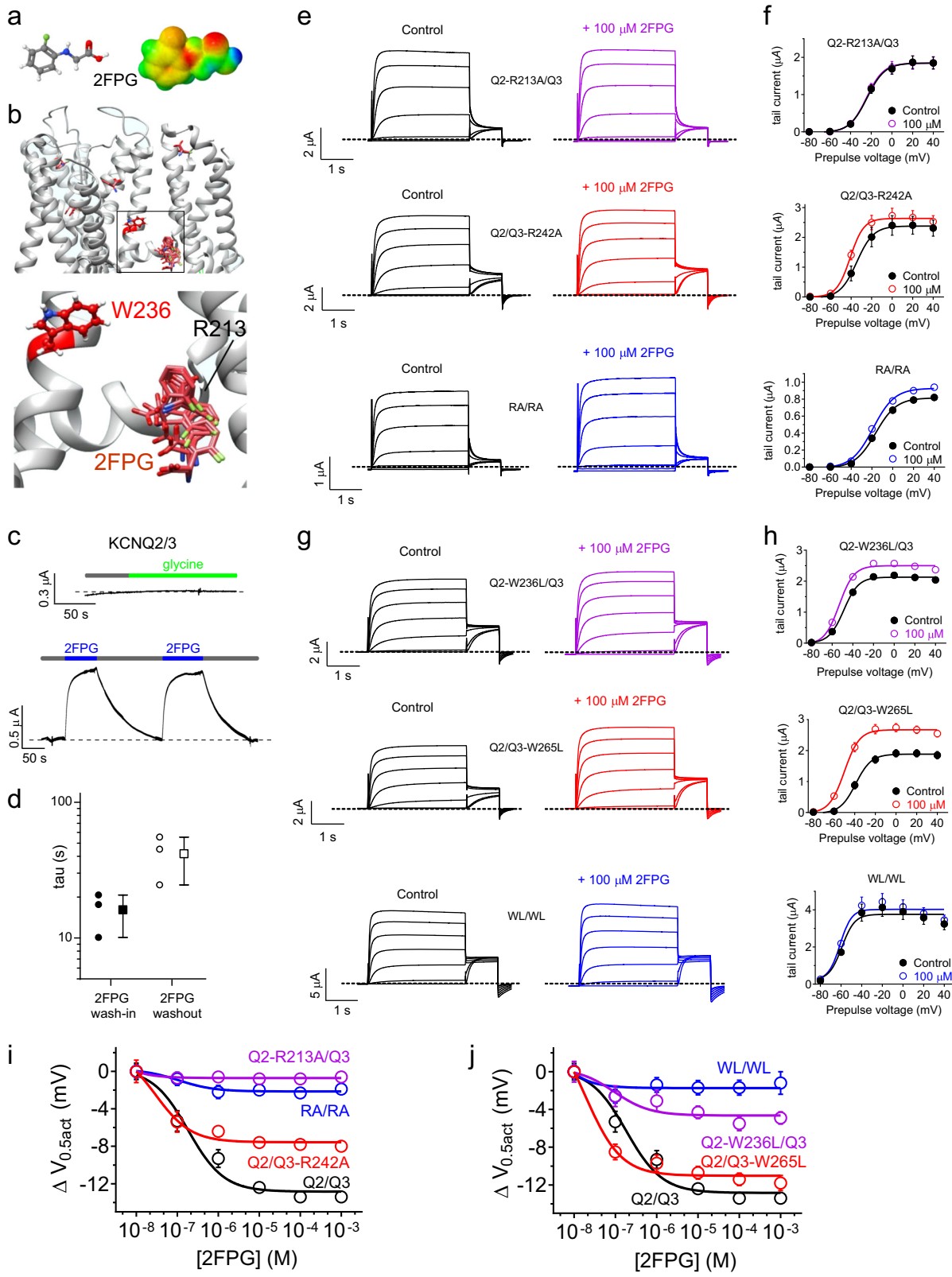

require several minutes to wash out. Instead, together with the docking predictions, the wash kinetics are more consistent with 2FPG accessing a deep binding site via the outer face of the membrane. We next tested the docking predictions in vitro using site-directed mutagenesis and TEVC. Mutating the S4-juxtaposed arginines in KCNQ2/3, we found that 2FPG required KCNQ2-R213 but not KCNQ3-R242 (Fig. 7e, f). Similarly, KCNQ2/

3 sensitivity to 2FPG was much more sensitive to mutation of KCNQ2-W236 than mutation of KCNQ3-W265 (Fig. 7g, h; dose responses in Fig. 7i, j). The dependence of 2FPG on the KCNQ2, but not KCNQ3, W and R residues in the previously discovered retigabine and GABA binding pocket[8,13] reflected data from the homomeric channels showing KCNQ2 but not KCNQ3 sensitivity to 2FPG (Fig. 4) (Supplementary Data 1, Tables 42–47).

**Fig. 7** 2FPG activation of KCNQ2/3 requires KCNQ2 R213 and W236. All error bars indicate SEM. **a** 2FPG structure and electrostatic surface potential map. **b** SwissDock result showing predicted binding of 2FPG to a chimeric KCNQ1-KCNQ3 model, with close-up of boxed region. **c** Representative trace showing effects at −60 mV on KCNQ2/3 current expressed in oocytes during wash-in and/or washout of 1 mM glycine or 100 μM 2FPG. **d** Mean time course of current increase and decrease during wash-in and washout respectively of 2FPG (100 μM) expressed as the tau of a single exponential function, quantified from traces as in panel (**c**), $n = 3$ oocytes (two wash-in/out cycles per oocyte, values averaged for each oocyte). **e**, **f** Mean traces (**e**) and tail current-voltage relationships (**f**) for wild-type and arginine-mutant KCNQ2/3 channels traces as indicated in the absence (Control) or presence of 100 μM 2FPG. RA/RA, KCNQ2-R213A/KCNQ3-R242A; $n = 5$. **g**, **h** Mean traces (**g**) and tail current-voltage relationships (**h**) for wild-type and tryptophan-mutant KCNQ2/3 channels traces as indicated in the absence (Control) or presence of 100 μM 2FPG. WL/WL, KCNQ2-W236L/KCNQ3-W265L; $n = 5$. **i** 2FPG dose responses of wild-type and arginine-mutant KCNQ2/3 channels as in e, f, quantified as shift in voltage dependence of activation ($\Delta V_{0.5act}$); $n = 5$. **j** 2FPG dose responses of wild-type and tryptophan-mutant KCNQ2/3 channels as in (**g**, **h**), quantified as shift in voltage dependence of activation ($\Delta V_{0.5act}$); $n = 5$

3FMSG also docked in silico between the S5 tryptophan (W265 in KCNQ3) and the arginine at the foot of S4 (R242 in KCNQ3) (Fig. 8a, b). Wash-in and wash-out experiments were again consistent with accessing a deep binding site from the extracellular face, with immediate onset of effects upon commencing wash-in or washout (Fig. 8c), and tau values of $15 \pm 3$ s for wash-in and $39 \pm 10$ s for washout (Fig. 8d). In contrast to data for 2FPG, but as expected from the 3FMSG sensitivity of KCNQ3 (Fig. 4), 3FMSG was more sensitive to mutation of KCNQ3-R242 than KCNQ2-R213, although mutating to alanine both the KCNQ2 and KCNQ3 equivalent arginines (RA/RA) produced a larger reduction in sensitivity (Fig. 8e, f). More definitively, 3FMSG sensitivity was dependent on KCNQ3-W265 but independent of KCNQ2-W236 (Fig. 8g, h; dose responses in Fig. 8i, j) (Supplementary Data 1, Tables 48–53).

To further validate these findings, we next compared GABA and glycine binding to KCNQ2/3 channels. As we previously found[8], tritiated GABA bound to wild-type KCNQ2/3. In contrast, tritiated glycine did not. Accordingly, cold glycine did not inhibit GABA binding to wild-type KCNQ2/3 (Supplementary Fig. 1a). Next, we found that the RA/RA mutation in KCNQ2/3 almost eliminated GABA binding (Supplementary Fig. 1b). Following the binding experiments with functional studies, we found that the RA/RA mutation eliminated activation of KCNQ2/3 by GABA, even at 1 mM (Supplementary Fig. 1c–e; (Supplementary Data 1, Table 54). These data support the validity of the in silico screening, which predicted that glycine would not bind (Fig. 2). In addition, the data expand our knowledge of the GABA binding site in KCNQ channels, indicating that in addition to the S5 Trp, the S4–5 Arg is also influential both in GABA binding and in the functional effects of GABA, as our data in Figs 7 and 8 suggest for 2FPG and 3FMSG. As GABA is thought to not cross the membrane (without active transport), the data lend weight to the idea that small molecules can access, from the extracellular face, a deep binding site within KCNQ channels that includes the S5 Trp and includes and/or is influenced by the S4-5 Arg.

We also studied the influence on 2FPG and 3FMSG effects of mutations in KCNQ2 and KCNQ3* homomers. Interestingly, KCNQ2-R213A channels exhibited increased 2FPG sensitivity compared to that of wild-type KCNQ2 (shifting the $EC_{50}$ fivefold, from $322 \pm 139$ nM to $57 \pm 20$ nM and marginally increasing the efficacy). This result is important as it provides further support for the idea that mutating the S4-5 Arg does not simply remove the ability of KCNQ2 activation to be modulated. Rather, it suggests contribution (either directly or indirectly) to a binding site that in a small-molecule specific manner can be damaged or augmented by mutating the S4-5 arginine to alanine. Conversely, KCNQ2-W236L channels were insensitive to 2FPG (Supplementary Fig. 2a–c; Supplementary Data 1, Tables 55, 56). KCNQ3* sensitivity to 3FMSG was reduced 44-fold ($EC_{50}$ shifted from $18 \pm 12$ to $787 \pm 80$ nM) by the W265L mutation, and maximal efficacy reduced by half (Supplementary Fig. 2d, e; Supplementary

Data 1, Table 57), reaffirming the importance of W265L in the effects of 3FMSG. KCNQ3*-R242 was nonfunctional and was awakened neither by 3FMSG nor retigabine (Supplementary Fig. 2f), a result that neither supports nor detracts from the premise of the S4-5 Arg influencing or participating in the small molecule binding pocket. Overall, the results further highlight the influence of the S5 Trp and the S4-5 Arg on modulation by 2FPG and 3FMSG.

**2FPG and 3FMSG KCNQ isoform selectivity arises primarily from functional selectivity.** To examine the mechanism of KCNQ isoform selectivity among glycine derivatives, we first assessed the combined effects of 2FPG and 3FMSG on homomeric KCNQ2 and KCNQ3 channels. A tenfold excess of 3FMSG (100 μM) subtly reduced the efficacy of 2FPG (10 μM) with respect to KCNQ2 activation by shifting the voltage dependence of 2FPG action such that efficacy at −40 mV was greatly reduced (although activation was still sufficient to hyperpolarize the oocyte membrane potential because effects were greater at more negative membrane potentials). In contrast, a tenfold excess of 2FPG (100 μM) did not alter the effects of 3FMSG (10 μM) on KCNQ3* activation at any membrane potential (Fig. 9a–d).

We next used a radioligand binding assay to quantify tritiated GABA binding to homomeric KCNQ2 and KCNQ3* channels expressed in oocytes. As we previously found[8], GABA bound to both KCNQ2 and KCNQ3 (Fig. 9e, f). Strikingly, 2FPG and 3FMSG (100 μM) were each able to compete out GABA binding to both KCNQ2 and KCNQ3*. Furthermore, while 3FMSG and 2FPG were equally able to outcompete GABA for KCNQ2 binding, 2FPG was not as effective as 3FMSG at outcompeting GABA for KCNQ3* binding. Together with the results in Fig. 9a–c, these data show that 2FPG and 3FMSG each bind to both KCNQ2 and KCNQ3*, to a site similar to or impinging (directly or allosterically) upon the GABA binding site. The data also suggest that while 2FPG has a higher binding affinity than 3FMSG for KCNQ2 (and the reverse is true for KCNQ3*), 3FMSG has a higher binding affinity for KCNQ2 than 2FPG does for KCNQ3*.

As each compound can bind to both KCNQ isoforms, isoform selectivity must arise predominantly from functional selectivity, not binding selectivity. There are clear binding preferences, otherwise a tenfold excess of the non-activating compound would greatly reduce the efficacy of the activating compound, and that did not occur (Fig. 9a–c). However, these preferences are not enough to explain the isoform selectivity shown in Fig. 4, i.e., a lack of functional effects of 2FPG on KCNQ3*, and 3FMSG on KCNQ2, even at high concentrations. Consistent with functional selectivity, using measurements of ion permeability series we found that 2FPG induces an increase in relative $Na^+$ permeability (compared to that of $K^+$) in KCNQ2 but not KCNQ3*, whereas 3FMSG induces a similar increase in relative $Na^+$ permeability of KCNQ3* but not KCNQ2 (Fig. 9g–i; Supplementary Data 1,

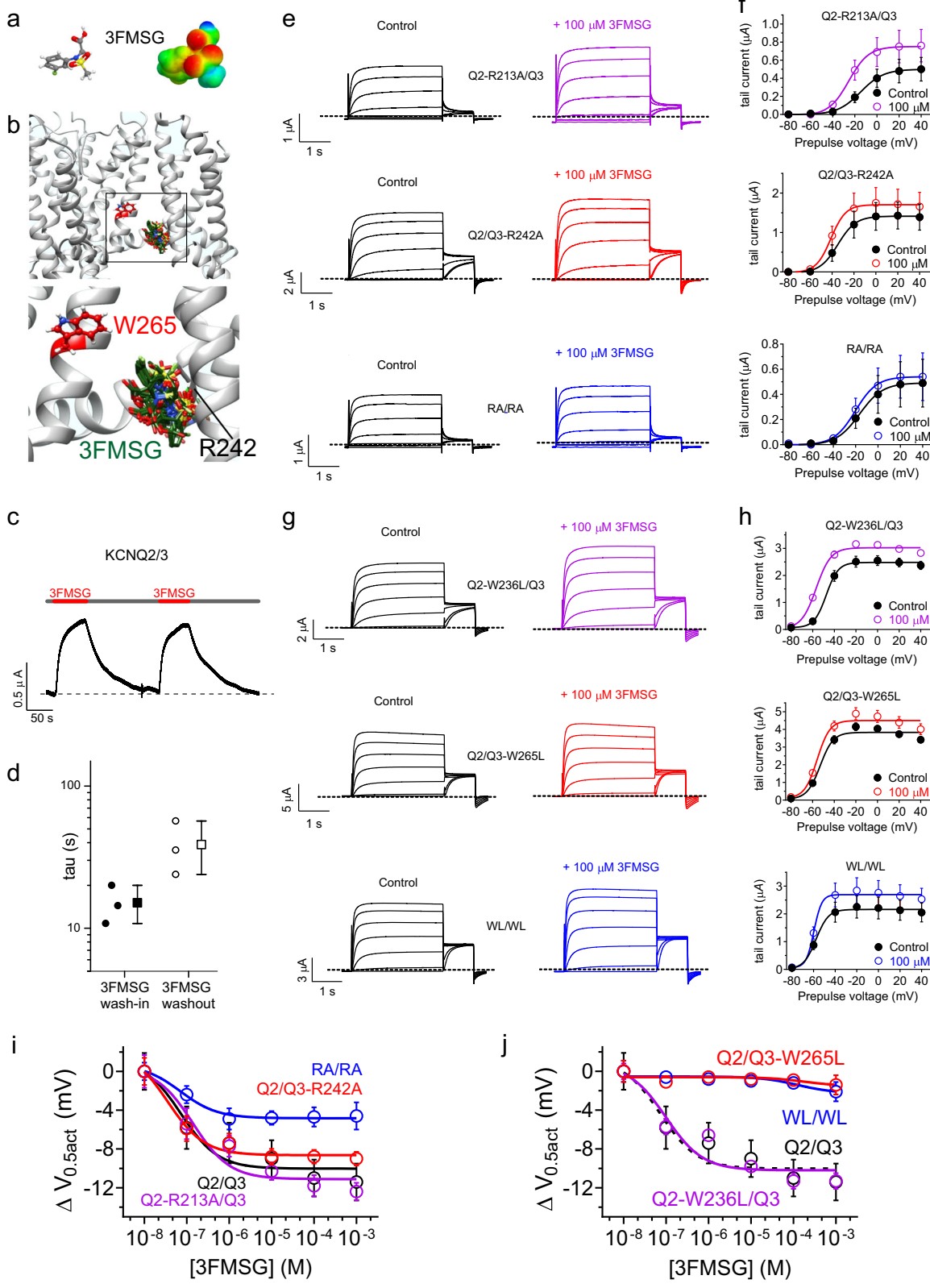

Tables 58, 59). The increase in the relative Na$^+$ permeability induced only by the correct compound/KCNQ isoform pairing suggested a conformational shift in the pore associated with channel activation. We previously observed a similar shift when KCNQ isoforms co-assembled via their pore domain with the SMIT1 *myo*-inositol transporter, which also negative-shifts the voltage dependence of KCNQ2 and KCNQ2/3 channels[18].

**2FPG and 3FMSG synergistically activate KCNQ2/3 channels.** As 2FPG and 3FMSG preferentially activate different KCNQ2/3 channel subunits (KCNQ2 and KCNQ3, respectively) we tested their ability to synergistically activate KCNQ2/3 by leveraging their isoform preferences. We used equal concentrations of each compound to avoid possible competition for the same-isoform binding site (see Fig. 9) and to instead leverage their binding

**Fig. 8** 3FMSG activation of KCNQ2/3 requires KCNQ3 R242 and W265. All error bars indicate SEM. **a** 3FMSG structure and electrostatic surface potential map. **b** SwissDock result showing predicted binding of 3FMSG to a chimeric KCNQ1-KCNQ3 model, with close-up of boxed region. **c** Representative trace showing effects at −60 mV on KCNQ2/3 current expressed in oocytes during wash-in and washout of 100 μM 3FMSG. **d** Mean time course of current increase and decrease during wash-in and washout respectively of 3FMSG (100 μM) expressed as the tau of a single exponential function, quantified from traces as in panel (**c**), $n = 3$ oocytes (two wash-in/out cycles per oocyte, values averaged for each oocyte). **e**, **f** Mean traces (**e**) and tail current-voltage relationships (**f**) for wild-type and arginine-mutant KCNQ2/3 channels traces as indicated in the absence (Ctrl) or presence of 100 μM 3FMSG. RA/RA, KCNQ2-R213A/KCNQ3-R242A; $n = 5$–6. **g**, **h** Mean traces (**g**) and tail current-voltage relationships (**h**) for wild-type and tryptophan-mutant KCNQ2/3 channels traces as indicated in the absence (Control) or presence of 100 μM 3FMSG. WL/WL, KCNQ2-W236L/KCNQ3-W265L; $n = 5$–6. **i** 3FMSG dose responses of wild-type and arginine-mutant KCNQ2/3 channels as in (**e**, **f**), quantified as shift in voltage dependence of activation ($\Delta V_{0.5act}$); $n = 5$–6. **j** 3FMSG dose responses of wild-type and tryptophan -mutant KCNQ2/3 channels as in (**g**, **h**), quantified as shift in voltage dependence of activation ($\Delta V_{0.5act}$); $n = 5$-6

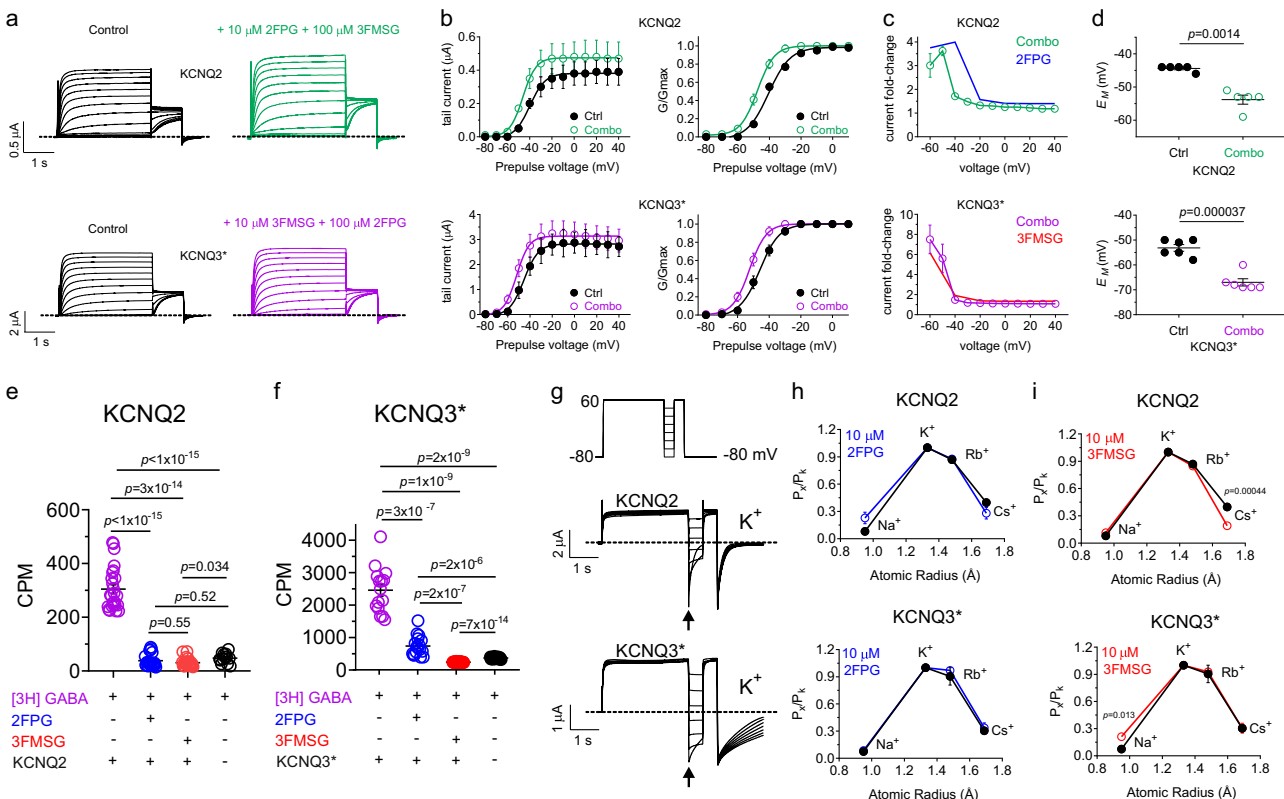

**Fig. 9** 2FPG and 3FMSG KCNQ isoform selectivity arises primarily from functional selectivity. All error bars indicate SEM. **a** Mean KCNQ2 and KCNQ3* traces in the absence (Control) versus presence of 2FPG + 3FMSG, concentrations as indicated ($n = 5$-6). **b** Mean KCNQ2 and KCNQ3* raw and normalized (G/Gmax) tail current versus prepulse voltages for traces as in panel (**a**) ($n = 5$-6). "Combo" indicates drug combinations shown with matching colors in panel (**a**). **c** Comparison of effects (expressed as fold-change in tail current versus prepulse voltage) of: 10 μM 2FPG alone (*blue*, from data as in Fig. 4) or in combination with 100 μM 3FMSG (*green*) on KCNQ2 current; 10 μM 3FMSG alone (*red*, from data as in Fig. 4) or in combination with 100 μM 2FPG (*purple*) on KCNQ3* current ($n = 5$-6). **d** Effects of the drug combinations as in panel (**a**) on the $E_M$ of unclamped oocytes expressing KCNQ2 or KCNQ3* ($n = 5$-6). **e** [3H]GABA binding quantified in counts per minute (CPM, measured over 30 min) to oocytes expressing KCNQ2 (or injected with water instead of KCNQ2 cRNA, as a control) in the absence or presence of 2FPG or 3FMSG (100 μM) as indicated; $n = 18$–25. Each point = 1 oocyte. **f** [3H]GABA binding quantified in counts per minute (CPM, measured over 30 min) to oocytes expressing KCNQ3* (or injected with water instead of KCNQ3* cRNA, as a control) in the absence or presence of 2FPG or 3FMSG (100 μM) as indicated; $n = 15$–30. Each point = 1 oocyte. **g** Exemplar traces showing KCNQ2 or KCNQ3* currents in response to the voltage protocol (inset) to quantify relative ion permeabilities; reversal potentials measured at arrow; K+ traces shown. **h** Effects of 2FPG (10 μM) on relative ion permeabilities of KCNQ2 and KCNQ3*, $n = 5$. **i** Effects of 3FMSG (10 μM) on relative ion permeabilities of KCNQ2 and KCNQ3*, $n = 5$

preferences (2FPG for KCNQ2; 3FMSG for KCNQ3). As discussed above, the binding preferences are not the primary mechanism underlying isoform selectivity of effects (selectivity that persists even at high concentrations), but based on data herein (Fig. 9a, b) would be predicted to permit synergy as they could favor KCNQ2 binding to 2FPG, and 3FMSG binding to KCNQ3, within KCNQ2/3 complexes. Accordingly, 2FPG and 3FMSG combined to potentiate KCNQ2/3 activity more than each compound alone (each at 10 μM) (Fig. 10a–d). Quantifying

current fold-increase (Fig. 10e) revealed that the drugs indeed synergistically activated KCNQ2/3 compared to either compound alone, also resulting in robust shifts in resting membrane potential (Fig. 10f), and robust speeding of KCNQ2/3 activation and slowing of deactivation (Fig. 10g) (Supplementary Data 1, Tables 60–62).

Investigating the synergy further, we dropped the concentrations to 1 μM each of 2FPG and 3FMSG and still observed synergy for KCNQ2/3 activation, increasing to a 40-fold increase

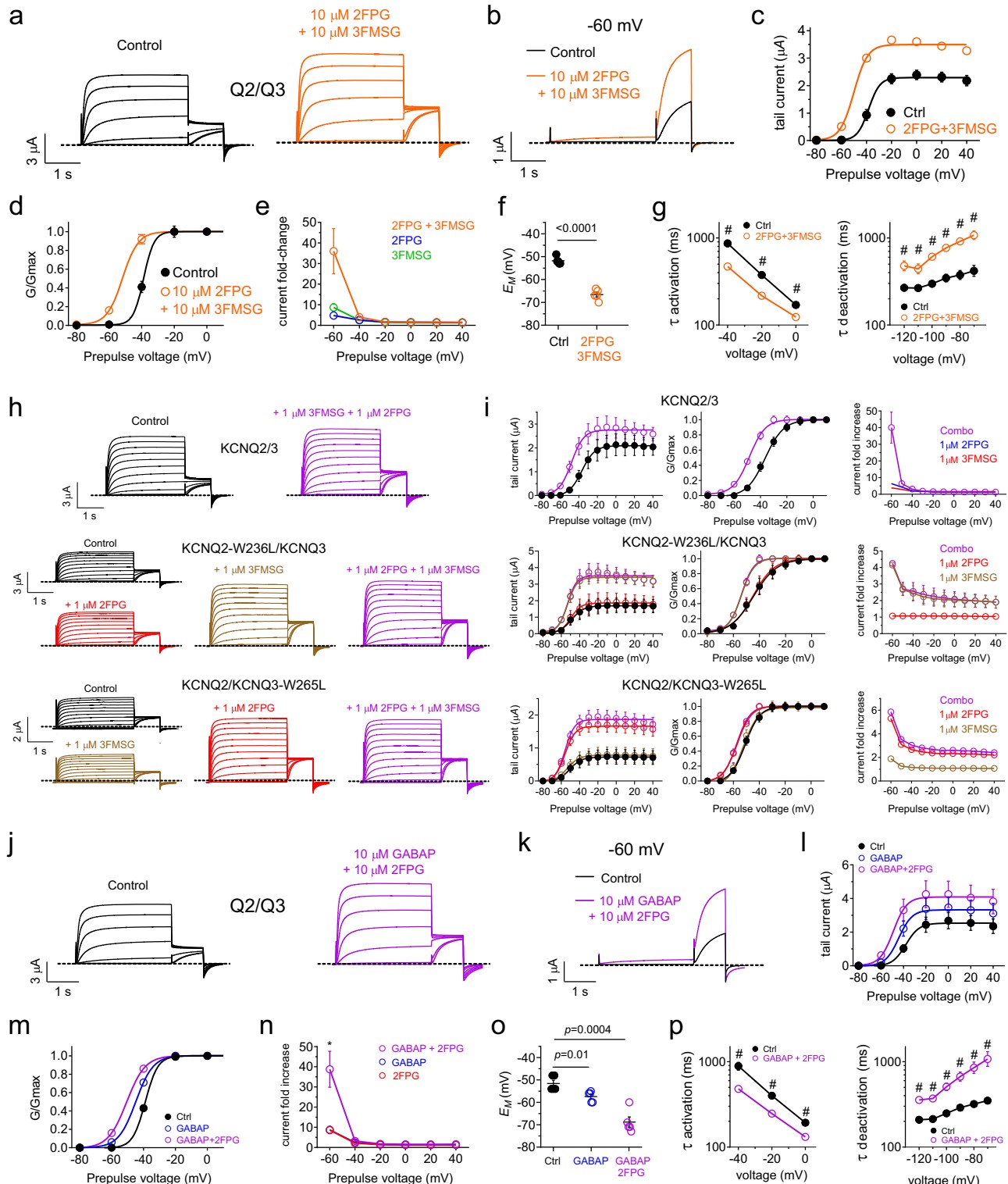

in current at −60 mV (Fig. 10h, i; Supplementary Data 1, Table 63), similar to the effects we had observed for 10 + 10 μM (Fig. 10e). We next analyzed the effects on synergy of binding-site mutations in either KCNQ2 or KCNQ3 within KCNQ2/3 channels. The KCNQ2-W236L mutation eliminated effects of 2FPG but not 3FMSG, and eliminated synergy between 2FPG and 3FMSG. Conversely, the KCNQ3-W265L mutation eliminated effects of 3FMSG but not 2FPG, and also eliminated synergy between 2FPG and 3FMSG (Fig. 10h, i; Supplementary Data 1, Tables 64, 65). The data suggest that synergy between 2FPG and

3FMSG arises from them each preferentially activating a different isoform within the KCNQ2/3 complex.

We previously found that gabapentin is a potent activator of KCNQ3 and KCNQ5 but not KCNQ2 channels, and that it also activates KCNQ2/3 heteromers[9]. Accordingly, we also found here that gabapentin synergizes with 2FPG with respect to KCNQ2/3 activation (Fig. 10j–m). The combination of gabapentin and 2FPG (10 μM each) produced a 40-fold increase in KCNQ2/3 current at −60 mV (Fig. 10n), a > 15 mV shift in resting membrane potential (Fig. 10o) and relatively strong speeding of

**Fig. 10** Leveraging the differential isoform preferences of 2FPG and 3FMSG for synergistic activation of KCNQ2/3. All error bars indicate SEM. **a, b** Mean KCNQ2/3 traces at −80 to +40 mV (**a**) or solely at −60 mV (**b**) in the absence (Control) versus presence of 2FPG + 3FMSG ($n = 5$). **c, d** Mean KCNQ2/3 tail current (**c**) and normalized tail currents (G/Gmax) (**d**) versus prepulse voltage in the absence (black) or presence (orange) of 2FPG + 3FMSG (each 10 μM) ($n = 5$). **e** Mean effect of 2FPG and 3FMSG (10 μM) alone or together on KCNQ2/3 current; $n = 5$. **f** Effect of 2FPG + 3FMSG (each 10 μM) on $E_M$ of unclamped oocytes expressing KCNQ2/3 ($n = 5$). **g** Effects of 2FPG + 3FMSG (each 10 μM) on KCNQ2/3 activation and deactivation rates; $n = 5$. # $P < 0.05$. **h** Mean wild-type ($n = 5$) or mutant ($n = 8$) KCNQ2/3 traces in the absence (Control) versus presence of 2FPG and 3FMSG (each 1 μM), separately or in combination. **i** Mean raw tail current, normalized tail current (G/Gmax), and current fold-increase versus prepulse voltage for channels indicated in the absence (black) or presence of 2FPG and 3FMSG (each 1 μM), separately or in combination; $n = 5–8$. Single-compound fold-effects for wild-type KCNQ2/3 from Fig. 4. **j, k** Mean KCNQ2/3 traces at −80 to +40 mV (**j**) or solely at −60 mV (**k**) in the absence (Control) versus presence of 2FPG + gabapentin (GABAP) ($n = 5$). **l, m** Mean KCNQ2/3 tail current (**l**) and normalized tail currents (G/Gmax) (**m**) versus prepulse voltage in the absence or presence of GABAP alone or in combination with 2FPG (each 10 μM) ($n = 5$). **n** Mean effect of 2FPG and GABAP (10 μM) alone or in combination on KCNQ2/3 current versus membrane potential; $n = 5$. **o** Effect of GABAP alone or with 2FPG (each 10 μM) on $E_M$ of unclamped oocytes expressing KCNQ2/3 ($n = 5$). **p** Effects of 2FPG + GABAP (each 10 μM) on KCNQ2/3 activation and deactivation rates; $n = 5$. # $P < 0.05$

activation and slowing of deactivation (Fig. 10p) (Supplementary Data 1, Tables 66–68).

Together with the data in Figs. 7–9 and Supplementary Figs 1 and 2 (showing a lack of synergy in homomeric KCNQ2 and KCNQ3* despite binding of 2FPG and 3FMSG to the neurotransmitter binding pockets of either isoform), the results in Fig. 10 demonstrate that combining KCNQ2- and KCNQ3-preferring compounds such as 2FPG and 3FMSG results in synergistic activation of KCNQ2/3. The data further demonstrate that the synergy arises because the combination of different isoform-preferring compounds leverages the heteromeric channel composition of KCNQ2/3 channels and the resultant mix of two different types of binding site.

## Discussion

Glycine and glutamate are structurally related to GABA, yet unlike GABA they do not exhibit negative electrostatic surface potential centered on the carbonyl group, an established property of several KCNQ channel openers that activate via KCNQ3-W265[11]. Here, we used mapping of electrostatic surface potential and docking to in silico-engineer a glycine derivative with predicted KCNQ-opening properties, with the initial hit (4FPG) resulting from the addition of a fluorophenyl group to the glycine amide group. Interestingly, 4FPG also activated KCNQ1, which lacks the S5 tryptophan required for activation by, e.g., retigabine and GABA, suggesting 4FPG can also activate via the S4/5-proximal arginine also important for KCNQ2/3 activation (although we did not pursue KCNQ1 mutagenesis studies herein). Remarkably, even subtle changes such as moving the fluorine atom two spaces along in the ring completely altered the KCNQ isoform selectivity of the glycine derivatives. While we do not yet understand the channel structural determinants underlying this selectivity switch, the finding suggests an avenue in which to explore future druggable derivatives that lack, e.g., KCNQ4 activity, as we observed for 2FPG and 3FMSG (Fig. 4).

We previously discovered that the heteromeric composition of KCNQ2/3 channels can be leveraged to potentiate the opening action of small molecules by combining two or more compounds with different KCNQ isoform preferences.[10] Here, we found that the principle holds for the glycine-based KCNQ activators, and also for the combination of KCNQ2-preferring 2FPG and gabapentin, a widely used analgesic that also exhibits anticonvulsant activity and which we previously found to isoform-selectively activate KCNQ3 and KCNQ5[9]. The KCNQ2/3 synergy approach may hold promise as a strategy for avoiding the individual toxicities of some compounds by combining them at lower (potentially safe) concentrations with compounds with alternate KCNQ isoform preferences, also at lower concentrations.

Interestingly, we also found that, similar to retigabine but not all KCNQ activators, the maximal efficacy of 2FPG and 3FMSG is

retained in heteromeric KCNQ2/3 channels, despite the insensitivity of KCNQ2 to 3FMSG and of KCNQ3 to 2FPG. This suggests either a dominance of effects of the drug-sensitive subunits within the complex, or may arise from the domain swapping nature of KCNQ channels endowing all four repeating units within the complex with drug sensitivity because each of the repeating units contains contributions from both KCNQ2 and KCNQ3.

With respect to the predicted deep binding site for the glycine derivatives, when we mutated the S4-5 arginine in the isoform that is sensitive to 2FPG (KCNQ2) versus 3FMSG (KCNQ3) to test the validity of the docking prediction, both in homomeric and heteromeric channels, we diminished or lost sensitivity and/or efficacy specifically to the respective drug, except for in one case in which the KCNQ2 R213A mutation increased sensitivity of homomers (and one mutant, KCNQ3*-R242A, was nonfunctional as a homomer). This suggests that the arginine residue either forms part of the binding site or impacts the way in which binding is translated into channel activation. This could possibly be because the drug binding disrupts interaction between the arginine and the cell membrane, or because mutating the arginine disrupts its interaction with the cell membrane. However, the arginine mutants do not greatly alter the voltage dependence of activation at baseline, suggesting against their mutation dramatically altering gating or voltage sensing per se at baseline, at least in KCNQ2/3 channels. Furthermore, we showed that KCNQ2/3 RA/RA mutant channels no longer bind GABA, and that 2FPG and 3FMSG displace GABA from wild-type KCNQ2 and KCNQ3* homomers. These support a model in which the S4-5 Arg is required for binding (either as part of the binding site or because it influences the conformation of the binding site) rather than solely for translating the effects of binding into activation.

In addition, the results of 2FPG and 3FMSG wash-in and washout studies are consistent with these molecules entering a deep binding pocket from the external face, and not having to first cross the cell membrane and then access the binding site from the inner face of the cell membrane. It is technically possible that one or both of the mutations induce allosteric effects that disrupt binding of GABA and/or glycine derivatives to a distant site, or that the S4-5 arginine influences the confirmation of the binding pocket but does not form part of it. The simplest explanation, however, remains that the S5 tryptophan and the S4-5 arginine line or contribute to the binding site.

There is one additional caveat. KCNQ2/3 α subunits are expected to exhibit domain-swapping, whereby the VSD of one subunit aligns with the pore module of the adjoining subunit. This may potentially result in mixed-isoform binding sites, which could complicate interpretation of results. This concern is allayed to a great extent by our findings for the homomeric mutant channels (Supplementary Fig. 2). While the results for the

homomers versus the heteromers indicate that neighboring sub-units certainly have an influence on the consequences of the mutations, especially in the case of the S4-5 Arg, the mutant homomeric channel data still support a role for the S4-5 Arg and the S5 W in mediating modulation by the glycine derivatives. Based on the data herein, we feel secure in stating that 2FPG and 3FMSG can each bind to both homomeric KCNQ2 and KCNQ3* channels, and that their isoform selectivity arises predominantly from the selectivity of their functional effects, and to a lesser extent their binding selectivity (the latter occurs, but cannot explain the lack of effects of 2FPG and 3FMSG on their non-preferred isoform even at high concentrations). Further, our data conclusively demonstrate that 2FPG and 3FMSG (or gabapentin) synergistically activate KCNQ2/3 channels by leveraging their isoform selectivity (both binding preference and functional selectivity) and the heteromeric composition of these channels.

The screening approach we used will be applicable to many other classes of small molecules with respect to predicting KCNQ channel activation, i.e., identify those compounds with the pre-ferred chemical properties, dock to filter out predicted non-binders and then validate in vitro. The docking program was able to correctly predict lack of glycine activity and also predicted binding of 2FPG, 4FPG and 3FMSG, but at this stage the model and/or docking program we use are not sophisticated enough to predict KCNQ isoform selectivity. While it is relatively trivial, once predicted KCNQ activity is identified, to test each of the KCNQ homomers for sensitivity in vitro, an accurate system in which isoform specificity could be used as a filter before in vitro screening, would be beneficial. With relatively few in silico screening steps and sufficient computing power, it may therefore be possible in the future to identify from massive commercially available chemical libraries safe, potent KCNQ openers that lack KCNQ4 opening activity.

## Methods

**Channel subunit cRNA preparation and Xenopus laevis oocyte injection**. We generated cRNA transcripts encoding human KCNQ1, KCNQ2, KCNQ3, KCNQ4, KCNQ5 or GLRA1 (NM_001146040) (GenScript, Piscataway, NJ, USA) by in vitro transcription using the T7 polymerase mMessage mMachine kit (Thermo Fisher Scientific), after vector linearization, from cDNA sub-cloned into plasmids incorporating Xenopus laevis β-globin 5′ and 3′ UTRs flanking the coding region to enhance translation and cRNA stability. We quantified cRNA by spectro-photometry. We generated mutant KCNQ2 and KCNQ3 cDNAs by site-directed mutagenesis using a QuikChange kit (Stratagene, San Diego, CA) and prepared the corresponding cRNAs as above. We injected defolliculated stage V and VI Xenopus laevis oocytes (Ecocyte Bioscience, Austin, TX and Xenoocyte, Dexter, MI) with KCNQ channel α subunit (5-20 ng) or GLRA1 (20 ng) cRNAs. We incubated the oocytes at 16 °C in Barth's saline solution (Ecocyte Bioscience) containing peni-cillin and streptomycin, with daily washing, for 2–5 days prior to two-electrode voltage-clamp (TEVC) recording.

**Two-electrode voltage clamp (TEVC)**. We performed TEVC at room tempera-ture using an OC-725C amplifier (Warner Instruments, Hamden, CT) and pClamp10 software (Molecular Devices, Sunnyvale, CA) 2–5 days after cRNA injection as described in the section above. For recording, oocytes were placed in a small-volume oocyte bath (Warner) and viewed them with a dissection microscope. Chemicals were sourced from Sigma, Matrix Scientific and Santa Cruz. (2-fluor-ophenyl) glycine, N-(3-fluorophenyl)-N-(methylsulfonyl) glycine and 2-(Tri-fluoromethyl)-DL-phenylglycine were each solubilized in bath solution at a stock concentration of 10 mM; 2-amino-2-(4fluorophenyl) acetic acid and 4-(tri-fluoromethyl)-L-phenylglycine were solubilized in 1 M hydrochloric acid at a stock concentration of 10 mM. All stock solutions were diluted in bath solution on the day of experiments. KCNQ2/3 channel activation was screened for using either 30 μM or 100 μM concentrations of each of the six compounds, then dose responses were conducted as appropriate. Bath solution was (in mM): 96 NaCl, 4 KCl, 1 MgCl₂, 1 CaCl₂, 10 HEPES (pH 7.6). Compounds were introduced into the oocyte recording bath by gravity perfusion at a constant flow of 1 ml per minute for 3 min prior to recording. Pipettes were of 1–2 MΩ resistance when filled with 3 M KCl. Currents were recorded in response to voltage pulses between −120 mV to −80 mV and + 40 mV at 20 mV intervals from a holding potential of −80 mV, to yield current-voltage relationships, current magnitude, and for quantifying activation rate. We analyzed data using Clampfit (Molecular Devices) and Graphpad Prism

software (GraphPad, San Diego, CA, USA); values are stated as mean ± SEM. We plotted raw or normalized tail currents versus prepulse voltage and fitted with a single Boltzmann function:

$$g = \frac{(A_1 - A_2)}{\left\{1 + \exp\left[(V_{\frac{1}{2}} - V)/V_S\right]\right\}} + A_2, \tag{1}$$

where $g$ is the normalized tail conductance, $A_1$ is the initial value at $-\infty$, $A_2$ is the final value at $+\infty$, $V_{1/2}$ is the half-maximal voltage of activation and $V_s$ the slope factor. Activation, deactivation, wash-in and washout kinetics were fitted with single exponential functions to yield a τ value.

**Relative permeability calculations**. According to the Goldman–Hodgkin–Katz (GHK) voltage equation:

$$E_{rev} = \frac{RT/F \ln(P_K[K^+]_O + P_{Na}[Na]_O + P_{Cl}[Cl]_i)}{(P_K[K^+]_i + P_{Na}[Na]_i + P_{Cl}[Cl]_o)}, \tag{2}$$

where $E_{rev}$ is the absolute reversal potential and $P$ is permeability. This permits calculation of the relative permeability of each ion if concentrations on either side of the membrane are known. A modified version of this equation was used here to determine relative permeability of two ions in a system in which only the extra-cellular ion concentration was known. Thus, relative permeability of $Rb^+$, $Cs^+$, and $Na^+$ compared to $K^+$ ions was calculated for KCNQ2 and KCNQ3* by plotting the $I/V$ relationships for each channel with each extracellular ion (100 mM) (using the voltage protocol shown in Fig. 9g) and comparing them to that with 100 mM extracellular $K^+$ ion to yield a change in reversal potential ($\Delta E_{rev}$) for each ion compared to that of $K^+$. Permeability ratios for each ion (X) compared to $K^+$ were then calculated as

$$\Delta E_{rev} = E_{rev,X} - E_{rev,K} = \ln\frac{P_X}{P_K}. \tag{3}$$

These values were then compared for each channel against $Rb^+$, $Cs^+$, $Na^+$, and $K^+$ containing 100 μM 2FPG or 100 μM 3FMSG and statistical significance was assessed using ANOVA.

**GABA and glycine radioligand binding studies**. Each group of oocytes was placed in a round-bottomed, 15-ml Falcon tube, washed with ND96, and then resuspended in ND96 containing 10 μM γ-[2,3-³H(N)]-aminobutyric acid (³H-GABA) or [2-³H]-Glycine (Perkin Elmer, Waltham, MA) at 25-45 Ci/mMol spe-cific activity (1 μM concentration) either alone, or with 100 μM 2FPG, 3FMSG or cold glycine for a 30 min incubation at room temperature. Oocytes were then washed four times in 16 °C ND96, transferred to individual wells in a 96 well plate and lysed in 0.2% SDS in ND96. Each oocyte lysate was transferred to a scintil-lation vial containing 5 ml Cytoscint scintillation cocktail fluid (MP Biomedicals, Santa Ana, CA). Vials were capped, shaken, and then allowed to sit at room temperature for at least 30 min before scintillation counting in a Beckmann Coulter LS6500 liquid scintillation counter.

**Chemical structures and silico docking**. We plotted and viewed chemical structures and electrostatic surface potentials using Jmol, an open-source Java viewer for chemical structures in 3D: http://jmol.org/. For in silico ligand docking predictions of binding to KCNQ2-5 channels, the Xenopus laevis KCNQ1 cryoEM structure (PDB 5VMS)[12] was first altered to incorporate KCNQ3/KCNQ5 residues known to be important for retigabine and ML-213 binding, and their immediate neighbors, followed by energy minimization as we previously described[8] using the GROMOS 43B1 force field[19] in DeepView[20]. We performed unguided docking of the compounds described in the manuscript, to predict potential binding sites, using SwissDock with CHARMM forcefields[21,22].

**Statistics and reproducibility**. All values are expressed as mean ± SEM. One-way ANOVA was applied for the majority of tests; if multiple comparisons were per-formed, a post hoc Tukey's HSD test was performed following ANOVA. Statistical significance was defined as $P < 0.05$. Oocyte experiments were each performed on at least two separate batches of oocytes to confirm reproducibility.

**Reporting summary**. Further information on research design is available in the Nature Research Reporting Summary linked to this article.

## Data availability

The raw datasets generated during the current study are available from the corresponding author on reasonable request. The source data underlying plots is presented in Supplementary Data 1.

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

## Acknowledgements

The authors are grateful to Angele De Silva (University of California, Irvine) for generating mutant channel constructs. This study was supported by the National Institutes of Health, National Institute of General Medical Sciences and National Institute of Neurological Disorders and Stroke (GM115189, GM130377 and NS107671 to GWA).

## Author contributions

R.W.M. performed the oocyte experiments and analyses, prepared most of the figure panels and edited the manuscript. G.W.A. conceived the study, selected compounds, performed in silico screening and structural analyses, wrote the manuscript and prepared the final figures.

## Competing interests

The authors declare no competing interests.
