## [Peer Review File · Communications Biology]

Reviewers' comments:

Reviewer #1 (Remarks to the Author):

In their study 'In silico re-engineering a neurotransmitter to isoform-selectively activate Kv channels', Manville and Abbott provide interesting data related to a rational approach for generation of compounds that activate various Kv7 channel subtypes. Based on prior findings suggesting the importance of a negative electrostatic surface potential, chemical modifications to glycine were screened in silico for desirable chemical properties, followed by functional screening of certain compounds. This led to the identification of three compounds (2FPG, 4FPG, 3FMSG) that act as Kv7 activators with apparently strong potency but modest efficacy. The drugs show many interesting properties, including unexpected subtype specificity (although the reasons are not yet known), and the ability for multiple drugs to interact in heteromeric channels containing subunits with different drug sensitivity. The data sets are comprehensive, and it is an interesting approach. Most of my comments are for clarification and possibly to include some more mechanistic discussions of the drugs.

Major Comments:

1. Although I don't see a need for the authors to collect additional data at this point, I would encourage a few changes to be made in future studies. I would suggest recording conductance-voltage relationships with more precision (e.g. smaller voltage intervals). In the present study they use 20 mV steps, and in several cases this did not seem adequate to correctly measure the steepness of voltage-dependent activation. Also, there may be subtle features of the voltage-dependence of activation that are missed with such wide intervals. How confident can you be in the extent of a modest (~10 mV) shift in voltage-dependence, when sampling in 20 mV intervals? More importantly, how can you confidently assess the valence/steepness of the conductance voltage relationship without more precise sampling?
2. I believe equation 1 is incorrect as written. It is missing a factor for valence (or steepness, slope...).
3. The manuscript could be improved with a clearer description of what the authors think is happening between residue W236/265 and R242/213? I can't speak for others, but I suspect a prevailing view in the field would be that R213 is possibly involved in a lipid interaction, rather than a direct interaction with the drug as the authors indicate. Could this be discussed in more detail? Beyond the docking approach is there strong evidence for binding at R242/213?
4. I had difficulty understanding the choice of drug concentrations for the synergism experiments. The concentrations used at least for 2FPG and 3FMSG are each quite high, almost saturating efficacy. It was not clearly described whether the combination of these drugs lead to enhancement of the maximal effect (gating shift, current increase) of either drug, or an increased sensitivity/potency? Could this be discussed further and choice of concentration justified?
5. I found it interesting and likely mechanistically relevant that the drug derivatives had nearly full efficacy in heterotetrameric channels with reduced numbers of drug-responsive subunit (eg. Fig. 6, 7).

Reviewer #2 (Remarks to the Author):

The study by Manville and Abbott tests whether they could develop glycine analogs to bind and activate specific isoforms of Kv7 channels using an *in silico* approach. A key goal of pharmaceutical companies is to develop compounds that activate Kv7.2/Kv7.3 heteromers but not Kv7.5 or Kv7.4 channels, as their activation could lead to adverse effects. In this paper, the authors describe several glycine analogs that can indeed activate Kv7.2 and Kv7.3 channels expressed in oocytes. They then show that these compounds might have anticonvulsant effects. This work is interesting but there are several major concerns.

1. Although this work is new, it lacks novelty. For instance based on the authors previous studies (Manville and Abbott Nat Comm. 2018; Manville and Abbott Mol Pharm. 2018, Manville et al Nat Comm. 2018) it not surprising that they could re-engineer glycine to bind to Kv7 channels. Overall this work is incremental.
2. Based on the supplementary tables (for instance see Supp. Table 1,2, 4,6) statistical significance of V05 values is only achieved when the data are normalized. This is an issue as there is no reason to expect V05 values to change when the data are normalized. Most investigators run statistics and reach their conclusions based on un-normalized G-Vs; as such, it is unclear whether the authors could reach their conclusions based on the current data.
3. The authors state that their new compounds have anticonvulsant properties. This is based on data presented in figure 4m. For the authors to reach this conclusion a much more thorough analysis is required. The authors need to provide a better description of the data and the experimental design. For instance, how exactly did they score the seizures? Also, they need to show a time course of seizure progression through the different seizure stages in animals treated with and without their glycine analogs.

Reviewer #3 (Remarks to the Author):

This is an interesting study to modify glycine to modulate KCNQ channels. The modification of glycine and binding of derivatives to KCNQ channels were based on *in silico* design and docking and supported by electrophysiology and mutagenesis experiments. A fluorophenyl ring to center glycine surface negative electrostatic potential on its carbonyl oxygen appears to be critical for the binding of derivatives to the channel. These studies provide insights on molecular interactions between agonists with KCNQ channels, providing a path for screening and rational design of compounds with therapeutic values for KCNQ-associated diseases. Some comments and questions are as follows.

1. Is glycine permeable to membrane? Could the lack of glycine effects on KCNQ channels be due to lack of membrane permeability? Some experimental evidence may help answering these questions.
2. Electrophysiological results: raw data of currents with and without compounds need to be shown in Figs 1, 2, 4, and 6.
3. Isoform selectivity of glycine derivatives is interesting. What structural differences of the KCNQ isoforms are responsible for such a selectivity?
4. Given the isoform selectivity (Fig 4) and synergy (Fig 8), experiments similar to those in Fig 6 and 7 should be performed on Q2 mutants and Q3 mutants, respectively, to avoid the complications derived from different contributions of KCNQ2 and KCNQ3 and their synergy.

Reviewer #4 (Remarks to the Author):

The authors previously reported a striking finding that an inhibitory neurotransmitter GABA activates KCNQ channel. This new paper handles a relevant topic, with a significantly expanded focus. After observing Glycine does not activate KCNQ channel, the authors tried re-engineering of effective derivative compounds by *in silico* screening, focusing on the presence of negative surface charge on the key carbonyl oxygen. After identifying promising candidates, they analyzed in detail the effect of 3 compounds (4FPG, 2FPG and 3FMSG) by electrophysiological analysis. They successfully confirmed the 3 compounds have activation effects on KCNQ channel, in a subunit selective manner.

I highly evaluate the scientific merit of this work in that it demonstrated an efficient and effective drug screening by analyses of chemical structure and docking *in silico*. The successful development of subunit selective activator of KCNQ is also of high impact, although the structural background for the subunit selectivity is not fully elucidated.

The manuscript is concise and written well, and the figures are friendly to readers. I have some comments which will benefit to improve the quality of the paper.

1. Glycine receptor, NMDA receptor

How about the effect of 2FPG, 4FPG and 3FMSG on Glycine receptor and NMDA receptor? No effect at all?

2. Figure 4

2FPG gives more effect on KCNQ2 than on KCNQ3, and 3FMSG gives more effect on Q3 than on Q2. Is it truly due to difference in the docking? It might be possible that e.g. 2FPG also binds to Q3, but does not induce further conformational change in favor of activation. It is worth examining if the 3FMSG competes with the 2FPG effect on Q2, and also if 2FPG competes with the 3FMSG effect on Q3, by competitive binding but with less activation effect.

3. Figure 2f, Figure 6b

Where does the carbonyl oxygen with a negative surface electrostatic potential locate? The relative allocation with R242 (R213) is not clear enough. Is the importance of the carbonyl oxygen for the docking fully supported? The reason why negative surface potential at the carbonyl oxygen is important, which is a key point of the *in silico* screening, is not clear from the point of view of structure. Also, how about the mutation effect of this Arg to amino acid residues other than Ala?

4. W265 (W236)

W265 (W236) looks to be far from the docked compounds and not at all involved in the binding. Any speculation why this Trp is critical? How about the mutation effect here to amino acid residues, other than Leu?

5. Figure 4m

Why error bars are missing here? Does 56% (black bar) mean e.g. 9 mice showed seizure out of 16 tested mice? If yes, the data number for this group is only one, and a statistical analysis is not possible. Please describe the detail of the data and statistical analysis here.

6. Seizure incidence analysis

Why the seizure incidence analysis was not performed after pretreatment by 2FPG (Figure 4), and also

by both 2FPG and 3FMSG (Figure 8)?

7. Figure 2g

How about the effect of 4FPG on KCNQ1?

8. Frog experiments

The description about the anesthesia of frogs for oocyte isolation would be necessary. Isn't it also necessary to describe for frog experiments that they follow the animal experiment guideline?

>The authors would like to thank the reviewers for their positive and constructive comments, which have helped us to further improve the manuscript. We have responded with many new experiments, including radioligand binding studies, permeability series experiments, and further synergy and mutagenesis studies. This resulted in additional figures and considerable edits to the text, which are highlighted in the manuscript and described point-by-point below.

Reviewers' comments:

Reviewer #1 (Remarks to the Author):

In their study 'In silico re-engineering a neurotransmitter to isoform-selectively activate Kv channels', Manville and Abbott provide interesting data related to a rational approach for generation of compounds that activate various Kv7 channel subtypes. Based on prior findings suggesting the importance of a negative electrostatic surface potential, chemical modifications to glycine were screened in silico for desirable chemical properties, followed by functional screening of certain compounds. This led to the identification of three compounds (2FPG, 4FPG, 3FMSG) that act as Kv7 activators with apparently strong potency but modest efficacy. The drugs show many interesting properties, including unexpected subtype specificity (although the reasons are not yet known), and the ability for multiple drugs to interact in heteromeric channels containing subunits with different drug sensitivity. The data sets are comprehensive, and it is an interesting approach. Most of my comments are for clarification and possibly to include some more mechanistic discussions of the drugs.

Major Comments:

1. Although I don't see a need for the authors to collect additional data at this point, I would encourage a few changes to be made in future studies. I would suggest recording conductance-voltage relationships with more precision (e.g. smaller voltage intervals). In the present study they use 20 mV steps, and in several cases this did not seem adequate to correctly measure the steepness of voltage-dependent activation. Also, there may be subtle features of the voltage-dependence of activation that are missed with such wide intervals. How confident can you be in the extent of a modest (~10 mV) shift in voltage-dependence, when sampling in 20 mV intervals? More importantly, how can you confidently assess the valence/steepness of the conductance voltage relationship without more precise sampling?

>We have taken this on board and used 10 mV steps for all the additional experiments we conducted for this study. We have also adopted this approach for other projects we are working on.

2. I believe equation 1 is incorrect as written. It is missing a factor for valence (or steepness, slope...).

>We have corrected, thank you.

3. The manuscript could be improved with a clearer description of what the authors think is happening between residue W236/265 and R242/213? I can't speak for others, but I suspect a prevailing view in the field would be that R213 is possibly involved in a lipid interaction, rather than a direct interaction

with the drug as the authors indicate. Could this be discussed in more detail? Beyond the docking approach is there strong evidence for binding at R242/213?

>The compounds dock close to the arginine and then when we mutate the arginine in the isoform that is sensitive to 2FPG (KCNQ2) versus 3FMSG (KCNQ3), we diminish or lose sensitivity specifically to the respective drug. This suggests that the arginine residue either forms part of the binding site or impacts the way in which binding is translated into channel activation. This could possibly be because the drug binding disrupts interaction between the arginine and the cell membrane, or that mutating the arginine disrupts its interaction with the cell membrane. However, the arginine mutants don't greatly alter the voltage dependence of activation at baseline. The simplest conclusion is that the compounds bind somewhere proximal to both the W and the R. However, we now include a version of the above discussion in the Discussion section (pages 17-18). We also performed additional studies to examine wash-in and washout (now Figures 7 and 8 panel c, d). These studies indicate a tau for wash-in of ~15 s, and washout of 40 s; in addition, increased channel activity begins to occur immediately upon wash-in, and begins to disappear immediately after onset of washout. These times and effects are inconsistent with 2FPG and 3FMSG crossing the cell membrane and integrating with the lipid bilayer from the inside, and instead are more consistent with accessing a somewhat deep binding pocket from the cell exterior, again consistent with the docking and mutagenesis data.

4. I had difficulty understanding the choice of drug concentrations for the synergism experiments. The concentrations used at least for 2FPG and 3FMSG are each quite high, almost saturating efficacy. It was not clearly described whether the combination of these drugs lead to enhancement of the maximal effect (gating shift, current increase) of either drug, or an increased sensitivity/potency? Could this be discussed further and choice of concentration justified?

>We have now performed additional synergy studies using 1 μ M (tenfold lower) and we get similar effects to with the higher dose (i.e., a 40-fold increase in current at -60 mV for the combination, compared to ~fivefold increase with either compound alone. The increase we observe is in efficacy (new Figure 10h, i). We also conducted further new experiments in which we tested the effects of 1 μ M 2FPG and 3FMSG alone or in combination on mutant KCNQ2/3 channels with a W-L substitution in either KCNQ isoform. These data nicely show that mutating the W in either isoform eliminates synergy completely, and maintains sensitivity only to the compound that prefers the other (wild-type) isoform (new Figure 10 h, i).

5. I found it interesting and likely mechanistically relevant that the drug derivatives had nearly full efficacy in heterotetrameric channels with reduced numbers of drug-responsive subunit (eg. Fig. 6, 7).

> Mutating the R or W of KCNQ2 in Q2/Q3 heteromers greatly reduced or eliminated efficacy of 2FPG; mutating the KCNQ3 R or W had much less effect on efficacy but KCNQ3 is relatively insensitive to 2FPG (see wild-type homomer data from Figure 4). Analogous results were seen for W mutants with 3FMSG, i.e., mutating the KCNQ3 W eliminated sensitivity/efficacy, while mutating the KCNQ2 W had no effect. The only result that did not go as expected from these experiments was that mutation of the KCNQ3 arginine had little effect on 3FMSG efficacy. This suggests that the KCNQ3 is more important for 3FMSG effects than is the arginine. We conducted additional studies to examine whether isoform selectivity arises from binding specificity or from specificity of activation itself. To do this we examined if cold 2FPG

or 3FMSG could compete out tritiated GABA binding. We found that either compound could compete out GABA binding to both KCNQ2 and KCNQ3 (Figure 9e, f). Thus, isoform selectivity of the compounds is less about binding selectivity (although there may be some preferences) and more about the isoform dependence of the effects of the compound once bound. We previously found a similar situation with GABA, which can bind to KCNQ2, 3, 4 and 5 (not KCNQ1, which lacks the S5 W) but only activates KCNQ3 and KCNQ5 (Manville et al., Nature Comms 2018).

Reviewer #2 (Remarks to the Author):

The study by Manville and Abbott tests whether they could develop glycine analogs to bind and activate specific isoforms of Kv7 channels using an in silico approach. A key goal of pharmaceutical companies is to develop compounds that activate Kv7.2/Kv7.3 heteromers but not Kv7.5 or Kv7.4 channels, as their activation could lead to adverse effects. In this paper, the authors describe several glycine analogs that can indeed activate Kv7.2 and Kv7.3 channels expressed in oocytes. They then show that these compounds might have anticonvulsant effects. This work is interesting but there are several major concerns.

1. Although this work is new, it lacks novelty. For instance based on the authors previous studies (Manville and Abbott Nat Comm. 2018; Manville and Abbott Mol Pharm. 2018, Manville et al Nat Comm. 2018) it not surprising that they could re-engineer glycine to bind to Kv7 channels. Overall this work is incremental.

>We disagree. In silico re-engineering a neurotransmitter to activate voltage-gated ion channels has never before been achieved. The fact that such minor changes to the structures of the neurotransmitters derivatives can completely alter KCNQ isoform selectivity is also novel and unexpected. To provide some perspective, there are currently 544 published papers on retigabine, the first-in-class Kv channel opening anticonvulsant, and papers on the mechanisms of action of retigabine continue to be published in highly ranked journals (19 papers on retigabine in 2019 so far; the first paper on retigabine was published in 1995). This is clearly, therefore, an area of much interest and our new findings bring a new angle to the field. We therefore contend that our new discovery is highly novel in the context of this field and current publishing expectations and norms.

2. Based on the supplementary tables (for instance see Supp. Table 1,2, 4,6) statistical significance of V05 values is only achieved when the data are normalized. This is an issue as there is no reason to expect V05 values to change when the data are normalized. Most investigators run statistics and reach their conclusions based on un-normalized G-Vs; as such, it is unclear whether the authors could reach their conclusions based on the current data.

>Thank you for pointing this out. We were fitting mean data in one dataset, in which case the fitting software provides statistics, while in the other dataset we were fitting each trace individually, getting a V0.5 value for each, then averaging and getting the SEM. We now use the latter approach and also stick to reporting the non-normalized GVs as suggested by the reviewer. It is also important to note that new statistical guidelines published in journals including Nature and The American Statistician strongly dissuade investigators from obsessing about attaining a P value less than 0.05, and instead encourage

reporting of P values but also taking into consideration effect sizes, which are generally more important. In the revised manuscript we mostly state the actual P values so that the reader can look at the effect size, consider the P value and judge the data, as we have done.

3. The authors state that their new compounds have anticonvulsant properties. This is based on data presented in figure 4m. For the authors to reach this conclusion a much more thorough analysis is required. The authors need to provide a better description of the data and the experimental design. For instance, how exactly did they score the seizures? Also, they need to show a time course of seizure progression through the different seizure stages in animals treated with and without their glycine analogs.

>We scored tonic seizure incidence as it is the most easily recognizable and severe seizure, in which the mouse stretches out its hind legs and remains prone for several minutes. The senior author (GWA) measured tonic seizure incidence and was blinded to whether the mice were injected with saline or test compound. We have used this system in several prior published studies. In response to the reviewer's comments, however, we have removed "anticonvulsant properties" from the abstract (even though the reduction in tonic seizures indicates anticonvulsant properties by definition) and report the finding in the results section as a reduction in tonic seizure incidence. There is such a large body of data in the manuscript on the molecular mechanisms of action of the new neurotransmitter derivatives that we felt a comprehensive seizure study can wait, as the main focus is on the mechanisms of K⁺ channel activation.

Reviewer #3 (Remarks to the Author):

This is an interesting study to modify glycine to modulate KCNQ channels. The modification of glycine and binding of derivatives to KCNQ channels were based on *in silico* design and docking and supported by electrophysiology and mutagenesis experiments. A fluorophenyl ring to center glycine surface negative electrostatic potential on its carbonyl oxygen appears to be critical for the binding of derivatives to the channel. These studies provide insights on molecular interactions between agonists with KCNQ channels, providing a path for screening and rational design of compounds with therapeutic values for KCNQ-associated diseases. Some comments and questions are as follows.

1. Is glycine permeable to membrane? Could the lack of glycine effects on KCNQ channels be due to lack of membrane permeability? Some experimental evidence may help answering these questions.

>Neither glycine nor GABA are membrane-permeable. GABA activates KCNQ channels (shown here and in a previous paper, Manville et al., *Nat Comms* 2018), so membrane permeability is not required to access the binding pocket used by GABA. To address the reviewer's question, we conducted 2 sets of experiments. First, we quantified the timing of channel activation and deactivation during wash-in and washout of 2FPG and 3FPG. Both compounds began to activate KCNQ2/3 channels immediately and also the effects began to reverse immediately upon washout (now Figures 7c, d and 8c, d). These kinetics are not compatible with the drugs having to cross the cell membrane to access the binding site, and instead point to extracellular access of a deep binding site as the docking studies predict and mutagenesis studies suggest. Second, we conducted radioligand binding competition studies (new Figure 9e, f) and

determined that 2FPG and 3FMSG can compete out GABA binding from homomeric KCNQ2 and KCNQ3 channels. These data confirm that 2FPG and 3FMSG share a similar binding pocket to GABA and that either molecule can bind to either channel isoform, i.e., as we previously found for GABA, isoform selectivity arises primarily from the isoform-specificity of the effects of binding, not from specificity of binding itself.

2. Electrophysiological results: raw data of currents with and without compounds need to be shown in Figs 1, 2, 4, and 6.

>We have now added these traces.

3. Isoform selectivity of glycine derivatives is interesting. What structural differences of the KCNQ isoforms are responsible for such a selectivity?

>This is an important question and the answer is that at the moment we do not yet know what endows the isoform selectivity. However, using new experiments (binding competition with tritiated GABA; Figure 9e, f) we have been able to ascertain that it is not the binding that is selective (2FPG and 3FMSG each bind to both KCNQ2 and KCNQ3), but rather the activating effects upon binding are isoform-specific. This explains how the compounds can be isoform selective despite binding to conserved residues. We also conducted new permeability series experiments and analysis and discovered that binding of 2FPG to KCNQ2, and 3FMSG to KCNQ3, but not vice versa, induces an increase in relative Na⁺ permeability indicative of a conformational shift in the pore (new Figure 9g-i). We previously observed this effect when we examined binding of the SMIT1 transporter protein to KCNQs, an event that also results in negative shifting of the voltage dependence of activation (Manville et al., 2017 Biophysical Journal). This provides further evidence that while the binding event is not isoform-specific, the downstream effects of binding are isoform-specific. We are currently embarking upon a new project to elucidate which aspects of the KCNQ channel architecture make each isoform selective for the functional effects of specific activators over others; we expect this could take a year or so as it will involve multiple mutagenesis scanning studies. It will be interesting to also study retigabine in this context, as despite a rich literature and almost 25 years of study on retigabine (544 papers and counting) it is also not yet known what endows some KCNQs with higher retigabine sensitivity versus others (e.g., KCNQ3 versus KCNQ2 – both are activated but KCNQ3 is more sensitive), other than the lack of sensitivity of KCNQ1 which arises from lack of the S5 W required for binding.

Reviewer #4 (Remarks to the Author):

The authors previously reported a striking finding that an inhibitory neurotransmitter GABA activates KCNQ channel. This new paper handles a relevant topic, with a significantly expanded focus. After observing Glycine does not activate KCNQ channel, the authors tried re-engineering of effective derivative compounds by in silico screening, focusing on the presence of negative surface charge on the key carbonyl oxygen. After identifying promising candidates, they analyzed in detail the effect of 3 compounds (4FPG, 2FPG and 32FMSG) by electrophysiological analysis. They successfully confirmed the

3 compounds have activation effects on KCNQ channel, in a subunit selective manner.

I highly evaluate the scientific merit of this work in that it demonstrated an efficient and effective drug screening by analyses of chemical structure and docking in silico. The successful development of subunit selective activator of KCNQ is also of high impact, although the structural background for the subunit selectivity is not fully elucidated.

The manuscript is concise and written well, and the figures are friendly to readers. I have some comments which will benefit to improve the quality of the paper.

1. Glycine receptor, NMDA receptor

How about the effect of 2FPG, 4FPG and 3FMSG on Glycine receptor and NMDA receptor? No effect at all?

>In response to this question, we were able to express glycine receptor GLRA1 and made some interesting findings (new Figure 5). Neither 2FPG nor 3FMSG activated GLRA1, but 3FMSG inhibited glycine-activated GLRA1 (35-40% inhibition at 100 μ M, compared to sub-micromolar EC50 values for KCNQ activation). We discuss the new data on pages 9-10.

2. Figure 4

2FPG gives more effect on KCNQ2 than on KCNQ3, and 3FMSG gives more effect on Q3 than on Q2. Is it truly due to difference in the docking? It might be possible that e.g. 2FPG also binds to Q3, but does not induce further conformational change in favor of activation. It is worth examining if the 3FMSG competes with the 2FPG effect on Q2, and also if 2FPG competes with the 3FMSG effect on Q3, by competitive binding but with less activation effect.

>We have conducted a series of experiments to answer this question (shown in new Figure 9). First we investigated whether a tenfold excess of 3FMSG could inhibit activation of KCNQ2 by 2FPG, and conversely whether a tenfold excess of 2FPG could inhibit KCNQ3 activation by 3FMSG. The latter had no effect but there was a subtle shift in the voltage dependence of KCNQ2 activation by 2FPG when 3FMSG was included, suggesting activation at -40 mV by 2FPG was slightly impaired (Figure 9a, b). From these studies, it was clear that there is no synergy between 2FPG and 3FMSG in homomeric KCNQ2 or KCNQ3 channels, and there was a suggestion of a minor inhibitory effect of 3FMSG on KCNQ2 activation by 2FPG.

>To scrutinize this further, we quantified the ability of 2FPG and 3FMSG to each compete with tritiated GABA for binding to homomeric KCNQ2 and KCNQ3 channels. We found that 2FPG and 3FMSG could each compete out GABA binding to both KCNQ2 and KCNQ3, demonstrating that either compound can bind to either isoform. In addition, 3FMSG was better at competing with GABA binding to KCNQ2 than was 2FPG at competing with GABA binding to 3FMSG, suggesting again that 3FMSG may be better at binding to KCNQ2 than 2FPG is at binding to 3FMSG (Figure 9e, f).

>To further understand how isoform activation selectivity might be achieved, we then conducted permeability series experiments and found that 2FPG increased the relative Na⁺ permeability of KCNQ2 but not KCNQ3, while 3FMSG increased the relative Na⁺ permeability of KCNQ3 but not KCNQ2 (Figure 9h, i). We observed a similar type of shift, indicative of a conformational shift in the pore module, when

SMIT1 co-assembled with KCNQ channels, which also resulted in a negative shift in the voltage dependence of activation (Manville et al., 2017 Biophysical Journal). We conclude that 2FPG and 3FMSG are isoform-selective primarily because despite each binding to both KCNQ2 and KCNQ3, there is isoform selectivity in their ability to induce a conformational shift associated with channel activation.

3. Figure 2f, Figure 6b

Where does the carbonyl oxygen with a negative surface electrostatic potential locate? The relative allocation with R242 (R213) is not clear enough. Is the importance of the carbonyl oxygen for the docking fully supported? The reason why negative surface potential at the carbonyl oxygen is important, which is a key point of the *in silico* screening, is not clear from the point of view of structure. Also, how about the mutation effect of this Arg to amino acid residues other than Ala?

> I have now added further docking images using a mesh surface for 4FPG and a sphere representation for the W, to give more of an indication of the predicted positioning and to show that there are two general predicted poses for 4FPG, one in which the carbonyl oxygen faces the Trp and the fluorine on the fluorophenyl ring faces the Arg, and vice versa (Figure 2g). We have not tried other mutations. Previous studies with the S5 Trp and retigabine showed that no other residue was tolerated at that position for retigabine to have its effects (Kim et al., 2015 Nature Comms). We currently cannot distinguish between these two orientations in our electrophysiological work, and it is also possible that both poses occur. For retigabine, the negative surface potential was required for interaction with the Trp.

4. W265 (W236)

W265 (W236) looks to be far from the docked compounds and not at all involved in the binding. Any speculation why this Trp is critical? How about the mutation effect here to amino acid residues, other than Leu?

> Previous studies with the Trp and retigabine showed that no other residue was tolerated at that position for retigabine to have its effects (Kim et al., 2015 Nature Comms) although for those studies activation but not binding per se was studied - in fact, it is rare for investigators in this field to truly quantify binding itself, as we have done here and in an earlier study (Manville et al., 2018 Nat Comms). A limitation of the docking is that it does not take into account changes in conformation that might bring those residues closer together, during voltage sensor activation, for example, or just inherent conformational flexibility. The tryptophan possibly can swing much closer to the arginine, and/or vice versa. In addition, when using a space-fill model for the side chains, it is apparent that there is not as much distance between the two residues as the ball and stick representation (that I used for clarity) suggests. I have now added further docking images using a mesh surface for 4FPG and a sphere representation for the W, to give more of an indication of the predicted positioning (Figure 2g). This aside, I speculate that the tryptophan might be involved in initial interaction with the compounds, bringing them into the pore and proximal to the putative deeper binding site closer to the arginine.

5. Figure 4m

Why error bars are missing here? Does 56% (black bar) mean e.g. 9 mice showed seizure out of 16 tested mice? If yes, the data number for this group is only one, and a statistical analysis is not possible. Please describe the detail of the data and statistical analysis here.

>We report the seizure incidence as a percentage (therefore no error bars) and use Fisher's exact test to compare the percent incidence between groups, yielding a P value. Apologies for not mentioning this in the Methods, this has now been added to the statistics section and to the figure legend (now Figure 4p).

6. Seizure incidence analysis

Why the seizure incidence analysis was not performed after pretreatment by 2FPG (Figure 4), and also by both 2FPG and 3FMSG (Figure 8)?

>The seizure incidence was not a major focus of the project but we included 3FMSG because we had initially tried that compound. In a future study we could examine the anticonvulsant actions of single and double compound combinations to optimize this.

7. Figure 2g

How about the effect of 4FPG on KCNQ1?

>We have now tried this and found that 4FPG activates KCNQ1, with similar efficacy and potency to 4FPG with KCNQ2, and with similar efficacy but higher potency for 3FMSG with KCNQ1 (in contrast, KCNQ1 is completely insensitive to 2FPG) (Figure 2 h-j). However, we found that while 4FPG both speeds KCNQ2 activation and slows its deactivation, for KCNQ1 there is no effect on activation rate, only deactivation rate (Figure 2k). This suggests that the S5 tryptophan, which is lacking in KCNQ1, may be important for governing effects on stabilizing the open state, but not destabilizing the open state, which is an interesting mechanistic possibility that we now discuss (paragraph 1, page 7).

8. Frog experiments

The description about the anesthesia of frogs for oocyte isolation would be necessary. Isn't it also necessary to describe for frog experiments that they follow the animal experiment guideline?

>We ordered pre-extracted, pre-digested oocytes directly from two companies (Ecocyte and Xenocyte) so we did not house frogs or perform surgery on them. Therefore we do not need an animal protocol for this aspect of the project.

Reviewers' comments:

Reviewer #1 (Remarks to the Author):

Thank you for addressing my comments. I felt that this was a comprehensive revision that addressed comments appropriately. The paper makes an interesting contribution to the field.

I just wanted to clarify point #5 (Reviewer #1) as my previous comment was opaque. What I meant to say was that I found it interesting that 2FPG (effective on KCNQ2 but not KCNQ3), or 3FMSG (effective on KCNQ3 but not KCNQ2) retain maximal efficacy in channels with mixed subunit composition (KCNQ2/KCNQ3). Retigabine seems to act this way, but certain other Kv7 activators like ICA-069673 appear to have intermediate effects in channels with mixed composition (of drug sensitive and insensitive subunits). This might be an interesting point to mention but that is just a suggestion.

The demonstration of binding with no effect for some of these drugs is an interesting addition to the paper.

Reviewer #2 (Remarks to the Author):

This is a revised manuscript that shows one can re-engineer glycine, an inhibitory neurotransmitter, to a KCNQ2/3 activator. Although I still consider that the findings are not surprising considering the previous published work by the authors, the authors have addressed most of my concerns.

As the authors state that seizures is not a major focus of the paper and prefer not to do additional experiments to strengthen that set of experiments I would suggest to remove them from the paper altogether (that is remove Figure 4p). The absence of the seizure experiments will not change their conclusions. Also, they do not provide any evidence that the anticonvulsant effects of the glycine analogs are due to KCNQ2/3 channel activation in vivo in the first place.

The authors state that "KCNQ4 activation is thought to lead to urinary retention by opening KCNQ4 in bladder detrusor muscle" (line 158-160). Recent work by Nelson and colleagues have shown that this not correct. The authors should revise that section of the paper and reference the following study:

The KV 7 channel activator retigabine suppresses mouse urinary bladder afferent nerve activity without affecting detrusor smooth muscle K⁺ channel currents.

Tyckocki NR, Heppner TJ, Dalsgaard T, Bonev AD, Nelson MT.
J Physiol. 2019 Feb;597(3):935-950.

Reviewer #3 (Remarks to the Author):

The authors provided additional experimental data to address comments and questions in the previous round of review, and these in general help clarify some of the questions. On the other hand, the assertion by the authors that the glycine derivatives reaches a deep pocket from outside of the membrane is not convincing. The putative pocket proposed in the manuscript is located in the intracellular side (N-terminus of the S4-S5 linker). There may be water filled crevice from intracellular side to expose this site but in the channel structure there is no obvious path from the extracellular side for a polar molecule to reach this site. The fact that the channels are Kv also makes it unlikely to

have a path in the channel protein for polar molecules across the membrane.

Since the entire manuscript is based on the foundation that the putative binding pocket selectively binds to glycine derivatives that contained a fluorophenyl ring concentrated negative surface potential at the native glycine carbonyl, it is important to verify if the putative binding pocket is indeed the key for observed phenomena. However, the results presented in the manuscript do not seem to show that with a comfortable certainty, as listed in the following.

1. As discussed above, channel structure does not seem to support that the putative pocket can be reached from extracellular side. The time course of the compounds effects may suggest that the binding site may be located in a different pocket that can be reached from extracellular side.
2. Docking to show binding pocket were done on the KCNQ3 homology model, but 4FPG and 2FPG had little effect on KCNQ3 function.
3. The S5 W apparently is not required for binding or effects since KCNQ1 is affected by 4FPG but does not have the S5 W.
4. While chemical specificity among glycine and derivatives is explained by their binding to the pocket, the subunit specificity, on the other hand, is explained by downstream effects of binding.

Based on these observations and interpretations, is it possible that the compounds bind to a different unknown pocket but the S4-S5 R and the S5 W are important for the downstream effects of binding? It seems that the binding pocket needs to be explored further. Some obvious tests come to mind. First, mutations of KCNQ2-R213 and KCNQ2-W236 should be tested in KCNQ2 homomeric channels. Both binding assays and functional assays should be performed to discern if the mutations alter binding or effects. Likewise, 3FMSG effects should be tested on homomeric KCNQ3 mutant channels. On the other hand, experimental data should be provided to show that glycine does not affect channel function because that the channels are unable to bind to glycine. In addition, it may be informative to test if the compounds can modulate the channel in the same way when allied from inside out patch.

Reviewer #4 (Remarks to the Author):

The authors revised the manuscript satisfactorily, fully responding to all points of my comments to the previous version. I have no more specific comments.

>The authors would again like to thank the reviewers for their comments, which have helped us to further improve the manuscript. We have responded with additional experiments, including further radioligand binding, electrophysiology and mutagenesis studies, leading to two additional multi-panel figures (new Figures 9 and 10). The additional data and edits to the text are highlighted in the “changes highlighted” manuscript file and described point-by-point below.

Reviewers' comments:

Reviewer #1 (Remarks to the Author):

Thank you for addressing my comments. I felt that this was a comprehensive revision that addressed comments appropriately. The paper makes an interesting contribution to the field.

I just wanted to clarify point #5 (Reviewer #1) as my previous comment was opaque. What I meant to say was that I found it interesting that 2FPG (effective on KCNQ2 but not KCNQ3), or 3FMSG (effective on KCNQ3 but not KCNQ2) retain maximal efficacy in channels with mixed subunit composition (KCNQ2/KCNQ3). Retigabine seems to act this way, but certain other Kv7 activators like ICA-069673 appear to have intermediate effects in channels with mixed composition (of drug sensitive and insensitive subunits). This might be an interesting point to mention but that is just a suggestion.

>We have added this interesting point to the Discussion (page 18, top).

The demonstration of binding with no effect for some of these drugs is an interesting addition to the paper.

Reviewer #2 (Remarks to the Author):

This is a revised manuscript that shows one can re-engineer glycine, an inhibitory neurotransmitter, to a KCNQ2/3 activator. Although I still consider that the findings are not surprising considering the previous published work by the authors, the authors have addressed most of my concerns.

As the authors state that seizures is not a major focus of the paper and prefer not to do additional experiments to strengthen that set of experiments I would suggest to remove them from the paper altogether (that is remove Figure 4p). The absence of the seizure experiments will not change their conclusions. Also, they do not provide any evidence that the anticonvulsant effects of the glycine analogs are due to KCNQ2/3 channel activation in vivo in the first place.

>We have removed Figure 4p as requested.

The authors state that "KCNQ4 activation is thought to lead to urinary retention by opening KCNQ4 in bladder detrusor muscle" (line 158-160). Recent work by Nelson and colleagues have shown that this not correct. The authors should revise that section of the paper and reference the following study:

The KV 7 channel activator retigabine suppresses mouse urinary bladder afferent nerve activity without

affecting detrusor smooth muscle K⁺ channel currents.
Tykocki NR, Heppner TJ, Dalsgaard T, Bonev AD, Nelson MT.
J Physiol. 2019 Feb;597(3):935-950.

>Thank you for this insight, we have removed specific discussion of KCNQ4 in the bladder as it no longer appears relevant to our study, given the new study from Tykocki et al.

Reviewer #3 (Remarks to the Author):

The authors provided additional experimental data to address comments and questions in the previous round of review, and these in general help clarify some of the questions. On the other hand, the assertion by the authors that the glycine derivatives reaches a deep pocket from outside of the membrane is not convincing. The putative pocket proposed in the manuscript is located in the intracellular side (N-terminus of the S4-S5 linker). There may be water filled crevice from intracellular side to expose this site but in the channel structure there is no obvious path from the extracellular side for a polar molecule to reach this site. The fact that the channels are Kv also makes it unlikely to have a path in the channel protein for polar molecules across the membrane.

Since the entire manuscript is based on the foundation that the putative binding pocket selectively binds to glycine derivatives that contained a fluorophenyl ring concentrated negative surface potential at the native glycine carbonyl, it is important to verify if the putative binding pocket is indeed the key for observed phenomena. However, the results presented in the manuscript do not seem to show that with a comfortable certainty, as listed in the following.

1. As discussed above, channel structure does not seem to support that the putative pocket can be reached from extracellular side. The time course of the compounds effects may suggest that the binding site may be located in a different pocket that can be reached from extracellular side.
2. Docking to show binding pocket were done on the KCNQ3 homology model, but 4FPG and 2FPG had little effect on KCNQ3 function.
3. The S5 W apparently is not required for binding or effects since KCNQ1 is affected by 4FPG but does not have the S5 W.
4. While chemical specificity among glycine and derivatives is explained by their binding to the pocket, the subunit specificity, on the other hand, is explained by downstream effects of binding.

Based on these observations and interpretations, is it possible that the compounds bind to a different unknown pocket but the S4-S5 R and the S5 W are important for the downstream effects of binding? It seems that the binding pocket needs to be explored further. Some obvious tests come to mind. First, mutations of KCNQ2-R213 and KCNQ2-W236 should be tested in KCNQ2 homomeric channels. Both binding assays and functional assays should be performed to discern if the mutations alter binding or effects. Likewise, 3FMSG effects should be tested on homomeric KCNQ3 mutant channels. On the other hand, experimental data should be provided to show that glycine does not affect channel function because that the channels are unable to bind to glycine. In addition, it may be informative to test if the compounds can modulate the channel in the same way when allied from inside out patch.

>To address these further points we embarked on a new series of experiments, as follows:

a) We now show that mutating the S4-S5 R in Q2/Q3 channels essentially eliminates binding of 3H-GABA (new Figure 9b). This further establishes the neurotransmitter binding pocket we reported last year [Manville et al., Nat Comms 2018] and indicates that the pocket incorporates/and or requires both the S5 W and the S4-5 R. There is a predicted path through to this pocket from the extracellular side, as we showed in the above referenced paper. GABA does not cross the membrane, so the new GABA binding data provide further evidence to support our hypothesis.

b) We now also show that mutating the S4-S5 R also eliminates activation by GABA (new Figure 9c-e). This backs up the results from the GABA binding study, described above. With these data in hand, the most likely conclusion is that the binding pocket incorporates the W and the R and can be accessed from the extracellular side. We have however added a new caveat that this could be an allosteric effect – this caveat of course applies to any protein mutation ever made, but we add it here (page 19, paragraph 2). We would also like to add that very few studies of small molecule effects on potassium channels have scrutinized binding per se to the extent that we have performed here. This especially applies to studies of anticonvulsants such as retigabine. Typically, authors are permitted to make their conclusions from functional effects alone. We agree that with the new data our study is now further improved and is much more rigorous than similar studies in this field that rely solely on functional outcomes and neglect to study binding directly. See also the next point.

c) We also examined glycine binding, in two new experiments. First, we show that 3H-glycine does not bind to wild-type Q2/Q3 channels. Second, we show that cold glycine does not prevent GABA binding to wild-type Q2/Q3 channels (new Figure 9a). These results further support our hypothesis that the fluorophenyl ring, and/or other elements that enhance negative electrostatic surface potential around the carbonyl group, are essential for binding of glycine derivatives; they also further validate the docking results and the entire premise of the in silico screening.

d) We could not study 2FPG or 3FMSG binding to the homomeric mutants because our assay is displacement of 3H-GABA binding, and that does not bind to the mutants. However, we already showed in the prior version of the manuscript that 2FPG and 3FMSG displace GABA from wild-type homomers (Figure 11e, f in the newly revised manuscript). Now that we include new data confirming that GABA does not bind to RA/RA-mutant KCNQ2/3 (and we previously showed that the W mutant inhibits GABA binding to KCNQ2/3 [Manville et al., 2018, Nature Comms]), the combination of these results leaves few if any other conclusions aside from the obvious ones that we describe herein.

e) We conducted the homomeric Q2 and Q3 mutant functional studies. Mutation of the KCNQ2 S5 W eliminated activation by 2FPG, whereas mutation of the S4-5 R enhanced activation by 2FPG. This observation of enhancement is a very important result. It gives us the first evidence that rather than just generically destroying the capacity for small molecule enhancement of activation, mutation of the S4-5 arginine can small-molecule specifically enhance potentiation in at least one case. This, together with the other new results, further supports our other extensive data that the R and W are influential in effects of 2FPG. For KCNQ3, mutation of the S4-5 R resulted in nonfunctional channels that could not be awakened by either 3FMSG or retigabine. Mutation of the KCNQ3*S5 W eliminated activation by 3FMSG, again supporting our hypothesis that the W is required for activation by 3FMSG. The results are shown in new Figure 10. Together with our extensive data on 2FPG and 3FMSG displacing GABA binding, our results showing that GABA binding is inhibited by mutation of either the W or the R, and the various

effects of mutating the W and the R in KCNQ2/3 heteromers, by far the simplest conclusion is that GABA, 2FPG and 3FMSG bind in the pocket we describe, that is formed by and/or heavily influenced by the S5 W and the S4-5 R.

Reviewer #4 (Remarks to the Author):

The authors revised the manuscript satisfactorily, fully responding to all points of my comments to the previous version. I have no more specific comments.

REVIEWERS' COMMENTS:

Reviewer #1 (Remarks to the Author):

I am satisfied with the paper and the authors' attention to comments.

Reviewer #2 (Remarks to the Author):

The authors have addressed my previous concerns.

Reviewer #3 (Remarks to the Author):

The authors satisfactorily responded my comments with extensive additional experiments.

Reviewer #4 (Remarks to the Author):

I checked the revisions in response to the comments of other reviewers. The revisions including new experimental data are full and satisfactory. I have no more specific comments and judge this manuscript with high scientific merits and strong impact is acceptable in the present form.